# Described Object Detection: Liberating Object Detection with Flexible Expressions

**Chi Xie**[1†]   **Zhao Zhang**[2†]   **Yixuan Wu**[3]   **Feng Zhu**[2]   **Rui Zhao**[2]   **Shuang Liang**[1*]

[1]Tongji University   [2]Sensetime Research   [3]Zhejiang University

chixie@tongji.edu.cn   zzhang@mail.nankai.edu.cn   shuangliang@tongji.edu.cn

## Abstract

Detecting objects based on language information is a popular task that includes Open-Vocabulary object Detection (OVD) and Referring Expression Comprehension (REC). In this paper, we advance them to a more practical setting called *Described Object Detection* (DOD) by expanding category names to flexible language expressions for OVD and overcoming the limitation of REC only grounding the pre-existing object. We establish the research foundation for DOD by constructing a *Description Detection Dataset* ($D^3$). This dataset features flexible language expressions, whether short category names or long descriptions, and annotating all described objects on all images without omission. By evaluating previous SOTA methods on $D^3$, we find some troublemakers that fail current REC, OVD, and bi-functional methods. REC methods struggle with confidence scores, rejecting negative instances, and multi-target scenarios, while OVD methods face constraints with long and complex descriptions. Recent bi-functional methods also do not work well on DOD due to their separated training procedures and inference strategies for REC and OVD tasks. Building upon the aforementioned findings, we propose a baseline that largely improves REC methods by reconstructing the training data and introducing a binary classification sub-task, outperforming existing methods. Data and code are available at this URL and related works are tracked in this repo.

## 1  Introduction

Detecting objects of interest within a scene using language is a pivotal area of focus. This field encompasses two key tasks: Open-Vocabulary object Detection (OVD) [10, 11, 20, 29, 47, 48] and Referring Expression Comprehension (REC) [21, 27, 24, 46, 52]. We present an intuitive illustration of these two settings in Fig. 1. The first task, OVD, expands the scope of object detection (OD) to any given short category name. However, these settings neglect the instances described by intricate descriptions. The second task, REC, focuses on spatially locating one target described by an expression and assumes the target must exist in the image. However, in real-world scenarios, if the described objects do not exist in the image, REC algorithms output false-positive results. Recent advancements have witnessed the joint training of bi-functional models, such as Grounding-DINO [25] and UNINEXT [44], which involve both OVD and REC data. Notwithstanding, these models still rely on separate training procedures and inference strategies for OVD and REC, and evaluate these two tasks independently.

As shown in Fig. 1, a more practical detection algorithm should be able to detect any described category, whether long or short, complex or simple, while discarding predictions in images where targets are absent. In order to address this significant yet often overlooked scenario, we propose the concept of **Described Object Detection (DOD)**. Note that this setting is a superset of OVD and REC.

---

[*]Corresponding author.
[†]Equal contribution.

37th Conference on Neural Information Processing Systems (NeurIPS 2023).

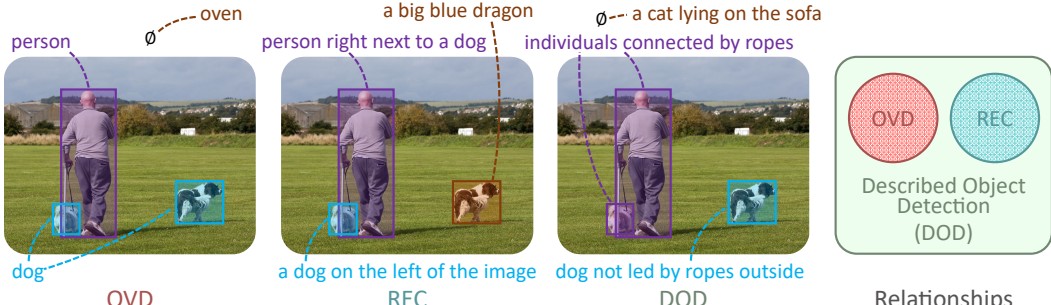

Figure 1: Examples showing the difference between REC, OVD and Described Object Detection (DOD). OVD detects arbitrary number (including zero, denoted with $\emptyset$) of objects based on a category name; REC grounds one region based on a language description, whether the object truly exists or not; DOD detect all instances on each image in the dataset, based on a flexible reference.

When the language expression is limited to a short category name, it becomes OVD. When we limit the images to detect objects known to be present in the images beforehand, it downgrades to REC.

Can the existing SOTA algorithms of the community support DOD tasks? To address this inquiry, this paper establishes the research foundation of DOD tasks by constructing a dataset, scrutinizing relevant methodologies, analyzing the relevant methods, and exploring improvement space.

**Motivation & real-world application of DOD.** OVD is limited to categorical detection, focusing on *classes* rather than specific attributes or relationships. It lacks detailed contextual understanding and cannot adapt to precise detection requirements from language. REC comprehend longer descriptions for attributes or relationships, but assumes the existence of one target in the image. This leads to false positives when the target is absent, limiting its practical usability. Consider detecting `individuals without helmets` on a construction site using camera data: OVD can detect `helmets` and `people` but not determine their relationship. REC locate one region in any image and generate false positives frequently. Existing solutions involve using separate models for object detection then relationship classification, or REC after image classification, both resulting in inefficiency.

Hence, there is a demand for language-based object detection: a model with strong generalization capabilities that can verify the existence of described objects in images and localize them based on arbitrary expressions. The proposed DOD task addresses this need and finds practical applications in: urban security, detecting `dogs without leashes` in communities, `clothes hung outdoors on streets`, `overloaded vehicles`, and `fallen trees on roadsides`; network security, like identifying sensitive images with violence or bloodshed within large datasets; (fine-grained) photo album retrieval based on descriptions or keywords; retrieval and filtering of web image data; specific event detection in autonomous driving, such as `pedestrians crossing the road`.

**Dataset & benchmark.** For DOD, we introduce the **Description Detection Dataset** ($D^3$, /dikju:b/), an evaluation-only benchmark containing 422 descriptions and 24,282 positive object-description pairs. Unlike previous OVD or REC datasets (see Fig. 2), $D^3$ stands out in three key aspects (see Tab. 1): 1) *Complete annotation*: All descriptions refer to objects annotated throughout the dataset, making $D^3$ a detection-style dataset akin to COCO [23]. 2) *Unrestricted description*: Annotations in $D^3$ include diverse and flexible language expressions, varying in length and complexity. 3) *Absence expression*: We include descriptions regarding absence of concepts, such as `a person without a safety helmet`, addressing an often-overlooked detection requirement. The details of $D^3$ is elaborated in Sec. 3. We evaluate state-of-the-art methods on $D^3$: OWL-ViT [29]/CORA [42] (OVD), OFA (REC) [39], and UNINEXT [44]/Grounding-DINO [25] (bi-functional) to provide a reference for the community. This benchmark may serve as a starting point for the DOD task.

**Findings & improvements.** The experimental analysis for different methods on $D^3$ yields some findings for future research (see Sec. 5): 1) Existing REC methods perform poorly, lacking confidence scores and the ability to reject negatives, and struggling with multi-target situations. This is due to their task formulation of grounding, i.e., matching between text and image region and not distinguishing positive and negatives. 2) OVD methods excel REC ones on DOD, though lengthy descriptions, which is not available in their training data, limit their performance. 3) Bi-functional methods, while superior to REC and OVD ones, share similar challenges with REC methods. Sometimes they are

Table 1: Comparison between the proposed dataset and previous REC datasets and OVD datasets.

| Dataset | annotation completeness | unrestricted description | absence expression | instance-level annotation |
|---------|------------------------|--------------------------|--------------------|---------------------------|
| RefCOCO | image-wise | ✓ | ✗ | ✓ |
| COCO | dataset-wise | ✗ | ✗ | ✓ |
| GRD | group-wise | ✓ | ✗ | ✗ |
| Ours | dataset-wise | ✓ | ✓ | ✓ |

surpassed by OVD models, indicating they have not fully benefited from REC and OVD. Based on these findings, we propose **a baseline OFA-DOD** that greatly improves a REC method, and outperforms current SOTAs. Its abilities to handle multiple targets and reject negative instances are improved by simple data reconstruction and an auxiliary sub-task. It is still far from a strong DOD method, but may provide some insights for research in the future.

## 2 Related Work

### 2.1 Relevant datasets and benchmarks

**Object detection datasets.** A variety of datasets have been proposed for object detection. Some have become standard benchmarks, like PASCAL VOC [9] and COCO [23]; while others are more frequently used for pretraining [35, 16, 2]. A few works have focused on special settings, such as LVIS [12] for long-tailed detection and ODinW [19] for zero-shot evaluation in the wild. Recently, V3Det [38]facilitates object detection with an extremely large vocabulary. Some are re-splitted and frequently used in OVD as well, like COCO and LVIS. As explained in Sec. 1, these datasets are all annotated with simple category labels rather than flexible language expressions like $D^3$.

**Referring expression comprehension datasets.** Several datasets have been introduced to evaluate REC methods, including RefClef [15], RefCOCO [46], RefCOCO+ [46], RefCOCOg [28], Visual Genome [17], and PhraseCut [40]. Some [15, 46] are collected interactively, and the expressions are more concise and less diverse. RefCOCOg is collected non-interactively, resulting in more complex expressions. Comparatively, Visual Genome focuses on visual relationships. All these datasets only annotate a few positive images for each category and leave other images unknown, which makes them unsuitable for the detection task.

**Other related tasks and datasets.** Several related tasks and benchmarks exist, but they differ significantly from DOD. Phrase Detection [31] lacks explicit negative labels as negative instances are unlabeled, and does not constitute a true detection task. Additionally, its references are simply phrases. In contrast, DOD ensures exhaustive annotation of positive and negative labels, and its references can be words, phrases, or sentences. Cops-Ref benchmark [6] focuses on evaluating the grounding capability of REC methods in difficult negative regions with related and distracting targets. It provides explicit negative certificates for only a limited set of images. In $D^3$, negative certificates are available across the entire dataset. Zero-shot grounding [34] centers on locating concepts not in the training set. It assumes the existence of the object referred by a reference in a image, and locates a single target per image, with a short phrase, while DOD makes no assumptions about the existence of the target, and locates zero to multiple targets, with varied expressions.

### 2.2 Current methods

**Open-vocabulary object detection methods.** Open-vocabulary detection is currently receiving increased attention. It aims to detect arbitrary classes using language for generalization, even when trained on a limited set of classes. The first approach, OVR-CNN [48], utilizes image-caption pairs for pretraining the visual encoder to enhance its zero-shot generalization capabilities. With the introduction of CLIP [33], models such as Detic [51], DetCLIP [45], RegionCLIP [50], and OV-DETR [47] have further advanced image and language embeddings pretrained using CLIP. ViLD [11] further distills knowledge from CLIP to inherit language semantics for recognizing novel classes. GLIP [20, 49] formulates object detection as a phrase grounding problem [32] and utilizes additional phrase grounding data to facilitate vision-language alignment.

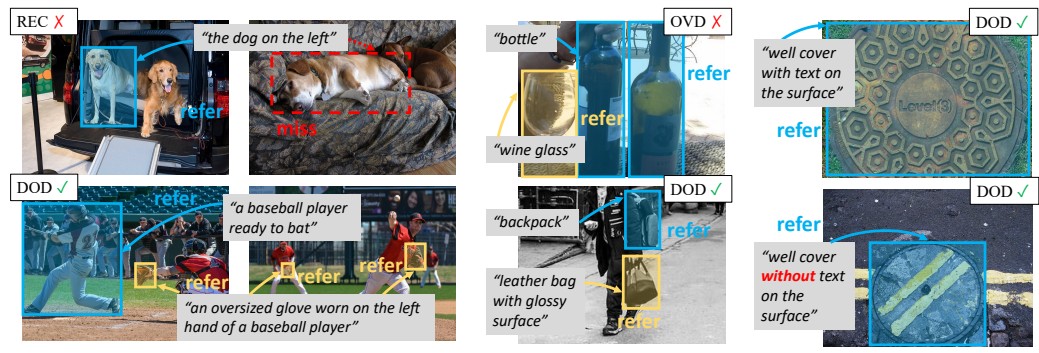

|(a) Complete annotation.|(b) Unrestricted reference.|(c) Absence expression.|

Figure 2: Some examples from previous datasets and the proposed $D^3$ dataset for DOD. (a) Our dataset for DOD is completely annotated for detection, while REC datasets like RefCOCO are not. (b) Our dataset has unrestricted reference, while OVD datasets like COCO are not. (c) Our dataset not only provides traditional presence descriptions, but also absence descriptions.

**Referring expression comprehension methods.** Existing works [7, 36, 21, 24, 39] can be divided into three categories. (1) Specialist models. Previously, two-staged works [13, 46] reformulate this as a ranking task. More recently, one-stage approaches [52, 37] speed up the inference process. (2) Multi-task models [53, 21, 24]. They usually design a unified formulation for a few closely related tasks. For example, SeqTR [53] unifies REC and RES (Referring Expression Segmentation) as a point prediction problem. (3) Multi-modal pre-training models [5, 26, 39]. Unified-IO [26] and OFA [39] propose unified sequence-to-sequence frameworks that can handle a variety of vision, language, and multi-modal tasks. Currently, OFA holds the SOTA among REC methods.

**Bi-functional models for REC and OVD/OD.** Some recent works [14, 8, 18, 25, 44] aim to handle tasks such as OVD (or OD) and REC concurrently within a single model. They typically restructure the training approach for these tasks, enabling a single model to learn from datasets related to both tasks. However, the inference process for each task remains distinct and independent of the other. FIBER [8] employs a two-stage pretraining strategy, separately utilizing image-text and image-text-box data to enhance data efficiency. More recently, Grounding-DINO [25] extends a closed-set detector by performing vision-language fusion at multiple stages and evaluating its performance on REC datasets. UNINEXT [44] reformulates various image and video tasks into a unified object discovery and retrieval paradigm. Despite these models sharing knowledge between detection and REC through pretraining, they are still treated as distinct tasks in these bi-functional models.

Methods with potential for DOD are continuously emerging and we will update them in this list.

## 3 Dataset

### 3.1 Dataset highlight

The proposed dataset is re-annotated on GRD [43], a dataset for RES [53, 24] and its variants [41, 22]. As briefly introduced in Sec. 1, it contains three major characteristics. In Fig. 2, we show some examples from previous datasets and $D^3$ to highlight them. Here we elaborate on them with a few other characteristics:

The first is *complete annotation*. For REC, the instances referred to by one description are only annotated in a few images. For other images without the annotation of this description, it is unknown whether the corresponding instance exists or not. That is to say, their annotations are not complete. Contrarily, as shown in Fig. 2a, in $D^3$, the objects referred to in all images by any description are annotated, as are the negative samples, like traditional object detection datasets.

The second is *unrestricted language description*. As shown in Fig. 2b, unlike (open vocabulary) object detection that retrieves objects with category names, we retrieve objects with language expressions, which is rather flexible. As is shown in Fig. 3d, the lengths of descriptions in $D^3$ vary a lot. The

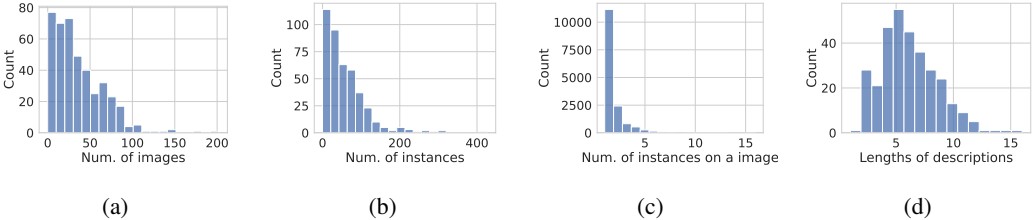

Figure 3: Distribution of (a) number of positive images for a description in the dataset, (b) number of positive instances for a description, (c) number of instances in a positive image for a description, and (d) lengths of descriptions.

shortest descriptions have one or two words, where the DOD task downgrades to OVD, while the longest may have 15 or more words, resulting in rather complex language expressions.

The third is *absence expression*. Current datasets with language description, like RefCOCO series for REC, usually describe objects with certain features. They usually focus on the ability to discover the existence of concepts but neglect their absence. Noticing the missing ability to verify such capability, we also annotate objects lacking a certain attribute. Fig. 2c shows an example with presence description and another with absence description from $D^3$. Such absence description makes up about one quarter of the references in this dataset. This is a first for existing benchmarks.

The fourth is *instance-level annotation*, a characteristic not held by GRD as it is intended for RES.

The fifth is *one description can refer to multiple instances* in an image, as in Fig. 3c. This is not true for REC datasets. If we regard category names as references, then OD datasets do have this feature.

In summary, the proposed dataset differs from the REC dataset primarily in terms of characteristics 1st, 3rd, and 5th. In contrast, when compared to OD datasets, the proposed dataset showcases disparities in the 2nd and 3rd characteristics, and when compared with GRD, in the 2nd, 3rd, 4th, and 5th characteristics. We refer the readers to the *supplementary materials* for more information about the characteristics of $D^3$ and more examples.

## 3.2 Annotation process

We utilize the GRD dataset [43] as the source for images, along with its original annotations. Originally, it is divided into multiple groups, each containing several references, with positive and negative samples annotated only within each group. We extend the annotations in three aspects:

**Adding instance-level annotation.** GRD is designed for RES, where each reference corresponds to one semantic mask across one image. However, for the DOD task, which requires the recognition and localization of individual instances, we annotate each instance referred to by a description with an individual bounding box (along with an instance mask). This is the basic step to adapt the dataset for instance localization.

**Adding complete annotations.** In addition to the intra-group annotation in GRD, we further annotate the positive and negative samples for each reference across the entire dataset. With complete dataset-wise annotations, the division into groups becomes unnecessary for evaluation, serving only as a means to organize references by scenarios. This enhancement makes the dataset suitable for detection tasks, significantly increasing the number of positive and negative samples.

Note that we use the complete annotation similar to COCO [23], i.e., explicit positive and negative certificates for all categories on all images, rather than federated annotation [12, 16]. This allows using mAP (mean Average Precision) as the evaluation metric, which is elaborated in Sec. 3.4.

**Adding annotations for absence expressions.** We have designed many absence descriptions based on the scenarios within the dataset, in addition to the traditional presence expressions in GRD. We annotate the instances in the images across the entire dataset with these absence expressions. This step increases the difficulty level of the proposed benchmark and enables the evaluation of existing models' ability to comprehend the absence of concepts.

We present a concise overview of the overall annotation process here. We organize groups of images and references (both for presence and absence). For each image, the references in its group are used. References from other groups may also appear, but with lower probability. We employ CLIP [33] to select a large number of candidates from these references in other groups. We manually check and adjust the hyper-parameters to make sure that such CLIP filtering usually do not miss positive refs. Subsequently, annotators select the positive references from these candidates (rather than from all references in the dataset) and add bounding boxes to the images. For more detailed information regarding the annotation process, please refer to *supplementary materials*.

## 3.3 Dataset statistics

**GRD statistics.** It has 10,578 images collected online, divided into 106 groups. Each group has around 100 images and 3 expressions referring to segmentation masks in this group, resulting in 316 references, 9,323 positive image-text pairs and 22,201 negative pairs. Note that it only annotates positive and negative samples inside each group, i.e., the annotation completeness is only **group-level**, so a reference will not be verified outside its group. The expressions have an average length of 5.9 words. We refer the reader to the original paper for specific statistics of GRD.

**$D^3$ statistics.** The proposed $D^3$ has 10,578 images, all from GRD. It has 422 well-designed expressions, including 316 expressions from GRD and 106 absence expressions we added (one for each scenario). The instance-level annotation results in 18,514 boxes.

Due to the effort in *complete annotation*, for a reference, each image in the dataset is annotated for possible positive and negative samples, i.e., the annotation completeness is **dataset-level**. Thus, there are 24,282 positive object-text pairs and 7,788,626 negative pairs, orders of magnitude larger than GRD. Among them, those with images and texts from the same scenario are probably more difficult, which includes 20,279 positive and 53,383 negative pairs. The average length of expressions is 6.3 words, due to the relative longer absence expressions. More statistics and examples of $D^3$ are available in *supplementary materials*.

## 3.4 Evaluation metrics

The classification of instances in $D^3$ is **multi-label**. Each description corresponds to a category. Naturally, there can be relationships between categories, such as parent-child hierarchies, synonyms, and partial overlap. When designing categories, we intentionally reduce parent-child or synonym relationships to ensure greater diversity and challenge. However, there exists partial overlap between categories. Therefore, in $D^3$, one instance may correspond to multiple descriptions, and the classification in $D^3$ is multi-label [12] rather than single-label [23], making it suitable for categories with relationships. An effective detector should assign all relevant positive categories (e.g., `dog not led by rope outside` and `clothed dog` for a clothed dog not led by rope outside) for an instance.

We use **standard mAP** for evaluation. Given the multi-label setting and the exhaustive annotation (all positive and negative labels are known for an instance) of $D^3$, category relationships will not affect the evaluation, so we can use consistent evaluation for each category across all images. We describe the evaluation process here. For inference, an instance predicted with category A and B is regarded as an instance for category A and an instance for B. The AP for each category is computed as follows: Predictions for each category across all images are sorted by score in descending order, and those with a ground truth IoU exceeding a threshold are counted as TP (and the ground truth is marked as taken), while the rest are counted as false positives. With these TP and FP instances, we calculate the precision, recall, and AP. The mAP is calculated by averaging the AP across all categories.

We use *FULL*, *PRES*, and *ABS* to denote evaluation on all descriptions, presence descriptions only, and absence descriptions only. If not noted explicitly, the *FULL* setting is adopted. The specific metrics for $D^3$ include: *Intra-scenario mAP:* For this metric, we perform evaluation on each image with only the descriptions from the image's scenario. The final metric is the mAP averaged on different IoU thresholds from 0.5 to 0.95, following COCO [23]. This is used as the default metric in our experimental settings. *Inter-scenario mAP:* It is similar to the intra-scenario mAP described above, except that for each image, we detect the possible instances with all 422 references. This is aligned with the common mAP in object detection datasets [23] and is much more challenging than the intra-scenario mAP.

# 4 Baselines

## 4.1 Existing baselines from different tasks

We choose multiple advanced methods to verify on $D^3$ from OVD, REC to bi-functional methods. More details of these methods and their inference process are in our *supplementary materials*.

**REC methods.** We employ the state-of-the-art REC method, OFA [39], with two variants. OFA is based on an encoder-decoder, sequence-to-sequence framework. It is a multi-modal multi-task generalist that deals with different tasks together and is trained on various tasks, including language tasks (masked language modeling), image-to-text tasks (image captioning and Visual Question Answering (VQA)), and localization tasks (REC). Notably, although it is trained with a detection dataset, it is not evaluated on object detection and achieves poor performance if we do. Currently, it holds the SOTA performance on standard REC benchmarks like the RefCOCO series.

**OVD methods.** We evaluate OWL-ViT [29] with two variants and CORA [42]. They are the SOTA methods on OVD tasks, with a vision transformer as well as a language transformer. They are pretrained with image-text contrastive learning and then fine-tuned on detection dataset.

**Bi-functional methods utilizing both REC and OVD data.** Methods falling into this category are not many but emerging fast recently. We test two methods: Grounding-DINO [25] and UNINEXT [44], each with two variants. Both of them are based on DETR [3]. They are pretrained on multiple datasets, including detection and REC datasets, and then evaluated with different strategies for different tasks.

## 4.2 A proposed baseline

$D^3$ is very challenging for existing works, as we will demonstrate in Sec. 5.1. We have selected one of these works for adjustment to provide a better baseline. The chosen work should (1) be capable of understanding text of various lengths; (2) excel in their original tasks; (3) have a framework with a rather simple technical design, allowing us to modify its components easily. We have chosen OFA because it (1) is a multi-modal multi-task framework with MLM (Masked Language Modeling) and image-to-text pretraining; (2) achieves SOTA on REC; (3) has a simple seq2seq framework.

However, OFA faces several problems that make it unsatisfactory for this task, as discussed in Sec. 5. First, forcing multiple tasks of different modalities into one seq2seq framework adversely affects the performance of specific tasks, especially tasks related to localization. Second, training on the grounding task results in poor ability to handle multiple instances. We evaluated the model on COCO detection, and it achieved less than 10 mAP. Thirdly, its REC paradigm also makes it predict only one instance, making it unable to reject negative images and irrelevant descriptions.

Therefore, we have made some modifications to OFA to make it more suitable for this task. The first modification is **granularity decomposition** to make it more suitable for localization. We have divided the pretraining tasks of OFA into two different granularities: global tasks (related to language modeling, such as captioning, VQA, MLM, etc.) and local tasks (related to localization, such as detection and REC). We have added an additional decoder parallel to the original decoder in OFA that handles the local tasks, while the original decoder focuses on the global tasks. This alleviates conflicts between different tasks and enhances localization.

The second modification is **reconstructed data** for pretraining on REC, aiming to improve multi-target localization. We have reconstructed the data for REC to ensure that (1) multiple references are input for an image, and (2) a reference does not necessarily correspond to one object, but zero or multiple. This results in a unified data format for detection and REC, although the labels may be noisy since they were not initially prepared for DOD.

The third modification is **task decomposition** to empower the model with the ability to reject false positives. We have reformulated the training on reconstructed data into two tasks: REC (for locating a region based on a reference) and VQA (for determining if a region and a reference match each other, essentially a binary classification). The second step is responsible for rejecting false positives.

We refer to the model with all three modifications as **OFA-DOD**. More details on the proposed improvements can be found in the *supplementary materials*. It is important to note that this model is far from perfect for the complex $D^3$ benchmark. As we will show in Sec. 5.1, although it outperforms existing methods, it serves as a baseline for future tasks on $D^3$.

Table 2: Comparison of different methods on the proposed dataset for different mAP metrics.

| Task | Method | Intra-scenario | | | Inter-scenario | | |
|------|--------|------|------|-----|------|------|-----|
| | | *FULL* | *PRES* | *ABS* | *FULL* | *PRES* | *ABS* |
| REC | OFA$_{base}$ | 3.4 | 3.0 | 4.3 | 0.1 | 0.1 | 0.1 |
| | OFA$_{large}$ | 4.2 | 4.1 | 4.6 | 0.1 | 0.1 | 0.1 |
| OVD | CORA$_{R50}$ | 6.2 | 6.7 | 5.0 | 2.0 | 2.2 | 1.3 |
| | OWL-ViT$_{base}$ | 8.6 | 8.5 | 8.8 | 3.2 | 3.7 | **4.7** |
| | OWL-ViT$_{large}$ | 9.6 | 10.7 | 6.4 | 2.5 | 2.9 | 2.1 |
| Bi-functional | UNINEXT$_{large}$ | 17.9 | 18.6 | 15.9 | 2.9 | 3.1 | 2.5 |
| | UNINEXT$_{huge}$ | 20.0 | 20.6 | 18.1 | 3.3 | 3.9 | 1.6 |
| | G-DINO$_{tiny}$ | 19.2 | 18.5 | 21.2 | 2.3 | 2.5 | 2.1 |
| | G-DINO$_{base}$ | 20.7 | 20.1 | **22.5** | 2.7 | 2.4 | 3.5 |
| DOD | OFA-DOD$_{base}$ | **21.6** | **23.7** | 15.4 | **5.7** | **6.9** | 2.3 |

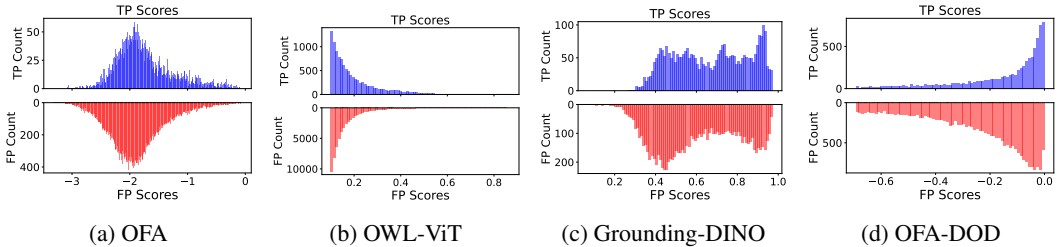

(a) OFA      (b) OWL-ViT      (c) Grounding-DINO      (d) OFA-DOD

Figure 4: Distribution of TP and FP scores from different baseline methods.

# 5 Experimental Analyses

## 5.1 Comparison of baselines on our metrics

We make comparisons on the baselines introduced in Sec. 4, mainly with the intra-scenario setting. Unless explicitly noted, this is the default setting, instead of the more difficult inter-scenario.

**Existing SOTAs are insufficient for DOD, and bi-functional models outperform others.** As demonstrated in Tab. 2, existing methods, while achieving SOTA performance on their original benchmarks, fall short in delivering strong performance on D$^3$. Among them, recent bi-functional methods [25, 44] are notably superior to others, and currently, OVD methods outperform REC. The inferiority of REC methods is likely due to their impractical setting described in Sec. 1, which involves predicting one and only one instance for each reference. We will delve into this further.

**Rejecting irrelevant references are difficult, which REC are naturally incapable of.** In contrast to intra-scenario evaluation, the inter-scenario setting assesses all references in the dataset for each image. Since references from other scenarios are likely not semantically relevant to the images, this necessitates the ability to reject irrelevant references for an image. This aligns with the evaluation in standard detection tasks. From Tab. 2, it is evident that OFA, a REC method, almost completely fails in this setting. This is caused by its prediction of a region for every reference, resulting in a large number of false positives when there are numerous candidate references. This underscores the importance of empowering REC methods with the ability to reject false positives. We find that none of the verified methods achieve good performance under the inter-scenario setting, indicating that existing methods are far from being capable of DOD. This highlights the challenge of D$^3$

**The proposed baseline outperforms existing methods.** The proposed baseline is based on OFA, but our improvements significantly enhance its performance. It outperforms all existing methods in the intra-scenario setting and surpasses them by a wider margin in the inter-scenario setting. This may suggest that the proposed baseline has a stronger ability to reject irrelevant references. Nonetheless, the proposed method is far from perfect and can only serve as a baseline for future research.

Table 3: Evaluation regarding different number of instances in a image for each reference.

| Method | No-instance FPPC (%) ↓ | One-instance mAP (%) ↑ | Multi-instance mAP(%) ↑ | | | |
|---|---|---|---|---|---|---|
| | | | 2 | 3 | 4 | 4+ |
| OFA | 100.0 | 14.8 | 9.5 | 7.9 | 5.4 | 3.7 |
| CORA | 17.3 | 9.7 | 8.4 | 9.5 | 9.0 | 8.5 |
| OWL-ViT | 41.9 | 21.1 | 17.3 | 16.6 | 16.0 | 14.0 |
| UNINEXT | 100.0 | 55.7 | 26.2 | 18.6 | 14.4 | 9.0 |
| G-DINO | 100.0 | 63.7 | 28.3 | 19.7 | 15.9 | 10.1 |
| OFA-DOD | 35.6 | 56.4 | 19.6 | 12.7 | 10.3 | 7.1 |

Table 4: Evaluation one references with various lengths.

| Method | *short* | *middle* | *long* | *very long* |
|---|---|---|---|---|
| OFA | 4.9 | 5.4 | 3.0 | 2.1 |
| OWL-ViT | 20.7 | 9.4 | 6.0 | 5.3 |
| UNINEXT | 18.5 | 23.3 | 17.4 | 16.1 |
| G-DINO | 22.6 | 22.5 | 18.9 | 16.5 |
| OFA-DOD | 23.6 | 22.6 | 20.5 | 18.4 |

## 5.2 Further analysis

**Absence descriptions are more difficult for most methods.** As shown in Tab. 2, the performance of baseline methods on *PRES* (presence descriptions) is consistently superior to that on *ABS* (absence descriptions). This suggests that existing methods may not effectively differentiate between the presence and absence of attributes in a language description.

**REC methods fail to provide good confidence scores.** We visualized the score distributions from baselines for TPs and FPs, to assess their capabilities in classification and confidence estimation. As in Fig. 4, the confidence scores from OFA do not exhibit a clear distinction between TP and FP cases. This can be attributed in part to the seq2seq framework in OFA, which does not directly yield confidence scores, and in part to the grounding formulation of REC, which identifies the image region most similar to the text description without distinguishing between positive and negative.

With a task decomposition step to enhance binary classification performance, our OFA-DOD demonstrates a significant disparity between TP and FP score distributions, yielding more reliable classification results. Note that this improvement does not necessitate modifications to the model framework or training datasets; rather, it is attributed to a more appropriate task formulation.

**Multi-instance detection is challenging for methods other than OVD.** For each image, $D^3$ can have zero to multiple instances **for a single description**. To assess how current methods handle varying numbers of instances, we conducted evaluations under three different settings: **no-instance**, where for a reference, evaluations are limited to images without any referred instance; **one-instance**, for images with a single instance; and **multi-instance**, for images with multiple instances. As shown in Tab. 3, OVD methods outperform others when multiple instances are referred by the description, although they may not be as competitive on the entire dataset or images with few instances. Notably, OWL-ViT maintains consistent performance even as the number of instances increases, which sets it apart from other methods. In contrast, REC and current bi-functional methods struggle in multi-instance scenarios. This highlights the strength of OVD methods in multi-target detection, while REC and current bi-functional approaches are less robust in such situations.

**REC and bi-functional methods lack the ability to reject negative instances.** In the **no-instance** column of Tab. 3, we do not report mAP since there are no positive instances in GT for the corresponding reference, making AP inapplicable. Predictions on such images are FPs, so we measure the ratio of images where FPs are produced to the total number of no-instance images for a given reference, namely False Positives Per Category (FPPC). We report the average FPPC over all references. We observe that most baselines are incapable of determining whether an image contains the referred target or not, yet they still produce predictions. This behavior is expected for REC methods. Bi-functional methods, trained and inferred with the REC task formulation, also exhibit this issue. Only the OVD method and our proposed baseline can effectively reject such negative image-text pairs.

Table 5: Ablation on the proposed baseline for its improvement components and the training data.

(a) Method components.

| OFA | GD | RD | TD | mAP(%) |
|-----|----|----|----|--------|
| ✓ | ✗ | ✗ | ✗ | 3.4 |
| ✓ | ✓ | ✗ | ✗ | 10.5 |
| ✓ | ✓ | ✓ | ✗ | 17.2 |
| ✓ | ✓ | ✓ | ✓ | 21.6 |

(b) Training data.

| REC | OD | I2T | MLM | mAP(%) |
|-----|----|----|-----|--------|
| ✓ | ✓ | ✓ | ✓ | 21.6 |
| ✓ | ✗ | ✓ | ✓ | 16.4 |
| ✓ | ✓ | ✗ | ✓ | 14.2 |
| ✓ | ✓ | ✓ | ✗ | 20.3 |

**OVD methods suffer from long descriptions greatly while others do not.** We partition the references according to their lengths and then evaluate on these partitions. The results are shown in Tab. 4, where *short*, *middle*, *long* and *very long* corresponding to references with 1~3, 4~6, 7~9, and more than 9 words. For *short* descriptions, which is close to OVD setting, OVD and bi-functional methods obtain similar performance. However, as the length of references increases, the performance of OVD methods decrease fast, while REC and bi-functional methods suffer less from this. We can see that OVD methods are sensitive to long references, as expected, while other two types do not.

More experiments and additional **qualitative results** are available in *supplementary materials*.

### 5.3 Ablation on the proposed baseline

**Method components.** In Tab. 5, we perform ablation on the proposed improvements in our baseline, step-by-step from OFA to OFA-DOD, to see how they affect the performance. Granularity decomposition (GD) makes the method more suitable for localization task. It disentangle tasks of global or local granularity by handling them with 2 separated branch. Reconstructed data (RD) uniforms REC and OD data into the same form, and prepares multi-instance samples with both short and long references. Task decomposition (TD) is proposed to help rejecting FPs. It breaks down the DOD task into a REC step followed by a VQA step. All three of them improve the performance obviously.

**Training tasks.** We also perform a drop-one-out ablation on the multi-modal multi-task training data, in Tab. 5b. **Detection** data provides samples for localization, especially multi-instance situation. It is instinctively important for learning to localize, and indeed matters for performance. **I2T** (image-to-text, like image captioning and visual question answering) often helps the generalization and zero-shot performance of multi-modal methods. We find that it does affect the zero-shot performance on $D^3$ greatly. **MLM** is theoretically important for language understanding and generalization. However, we find it actually is not. Removing the MLM task has no significant effect on the performance. We surmise that the generalization ability of OFA-DOD on $D^3$ mainly comes from I2T.

## 6 Conclusion and Limitation

In this paper, we bring the Described Object Detection (DOD) task to the foreground. For this task, we introduce a dataset called $D^3$, which annotates described objects without omission and features flexible language expressions, whether long or short, complex or simple. Our evaluation of SOTA methods from REC or OVD on $D^3$ reveals challenges faced by REC, OVD, and bi-functional approaches. Based on these observations, we propose a baseline that largely improves REC methods for DOD task. We believe that the dataset and findings will contribute to advancing the understanding and development of DOD methods, facilitating future research in this area.

**Limitation and broader impact.** This work does have some limitations. Due to the significant annotation cost brought by our complete annotation process, we are unable to propose a huge dataset with millions or billions of images. Besides, the evaluation and findings in this work may be dependent on the choice of descriptions and the image sources. This work only serves as a starting point for DOD and we hope there will be other DOD datasets with larger scales. In the broader community, compared to traditional detection algorithms, DOD models have a lower customization threshold, enabling users to specify the detection target using language. This may lead to potential abuse.

**Future work.** During peer-review process, some new works with potential for DOD emerges, including Shikra [4], Kosmos-2 [30] and Qwen-VL [1]. We will continue to investigate such methods for DOD and update them in this list.

**Acknowledgments.** This work was supported in part by the National Natural Science Foundation of China under Grant 62076183, 61936014 and 61976159, in part by the Natural Science Foundation of Shanghai under Grant 20ZR1473500, in part by the Shanghai Science and Technology Innovation Action Project under Grant 20511100700 and 22511105300, in part by the Shanghai Municipal Science and Technology Major Project under Grant 2021SHZDZX0100, and in part by Hetao Shenzhen-Hong Kong Science and Technology Innovation Cooperation Zone (HZQB-KCZYZ-2021045). The authors would also like to thank the anonymous reviewers for their careful work and valuable suggestions.

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
