# Described Object Detection: Liberating Object Detection with Flexible Expressions

*——— Supplemental File ———*

Chi Xie[1†]    Zhao Zhang[2†]    Yixuan Wu[3]    Feng Zhu[2]    Rui Zhao[2]    Shuang Liang[1*]

[1]Tongji University    [2]Sensetime Research    [3]Zhejiang University

chixie@tongji.edu.cn    zzhang@mail.nankai.edu.cn    shuangliang@tongji.edu.cn

## Abstract

**Content.** In this supplemental file, we provide more details to support the paper.

▶ **Dataset details and more examples** for the proposed dataset, $D^3$, are presented in Sec. 1.

▶ **Evaluation of previous methods** are presented in Sec. 2, which describes the existing works we evaluated and the specific details regarding how we adapt them to the DOD task.

▶ **Details of the proposed baseline** are presented in Sec. 3.

▶ **More experimental results** are shown in Sec. 4, including both quantitative and qualitative results.

## 1 Dataset Details

### 1.1 More examples

In Section 3.1 of the main paper, we introduced the key features of the proposed $D^3$ dataset. Here we provide further examples to support this part.

**Complete annotation.** The first characteristic of $D^3$ is the dataset-level complete and thorough annotations, setting it apart from REC datasets [20, 9]. In $D^3$, every image is annotated for possible positive and negative instances, as demonstrated in Fig. 1. This figure includes several images with positive instance labels (first row) and several images with negative instance labels (second row) for each of the four descriptions. Such comprehensive annotation makes the proposed dataset well-suited for detection tasks.

In comparison, REC datasets like RefCOCO [20, 9] only annotate several positive instances in a few images for each description, leaving all the other images without annotations for that particular description; thus, their annotation completeness is limited to the image-level. On the other hand, GRD [18] annotates a description for a group of images while dividing the entire set into multiple groups, resulting in an annotation completeness at the group-level.

**Unrestricted description.** The categories in $D^3$ encompass more than just simple object names, such as `cat`, `dog` and `bird` found in typical object detection datasets [6, 2, 13]. As illustrated in Fig. 2, the descriptions are expressed in unrestricted natural language. The longer and more complex descriptions resemble references found in REC datasets [20, 9, 4]. For instance, a description like `a fisher who stands on the shore and whose lower body is not submerged by water` comprises 16 words and encompasses multiple attributes like `fisher`, `stands on the shore` and `lower body is not submerged by water`. These attributes are semantically abstract and visually diverse. On the other hand, the shorter and simpler descriptions can be similar to the category names in OD datasets, such as `backpack`, `swing bench` and `a sailboat`. This illustrates that the descriptions of objects in $D^3$ are free-form and unrestricted, covering a wide range of description types present in both REC and OD datasets.

**Absence description.** To the best of our knowledge, the proposed dataset is the first annotated dataset specifically designed to address absence descriptions. Examples with annotations for both presence and absence descriptions from our dataset ($D^3$) are illustrated in Fig. 3. For visualization purposes, we have selected some absence descriptions that have contradictory presence descriptions. The

---

*Corresponding author.

†Equal contribution.

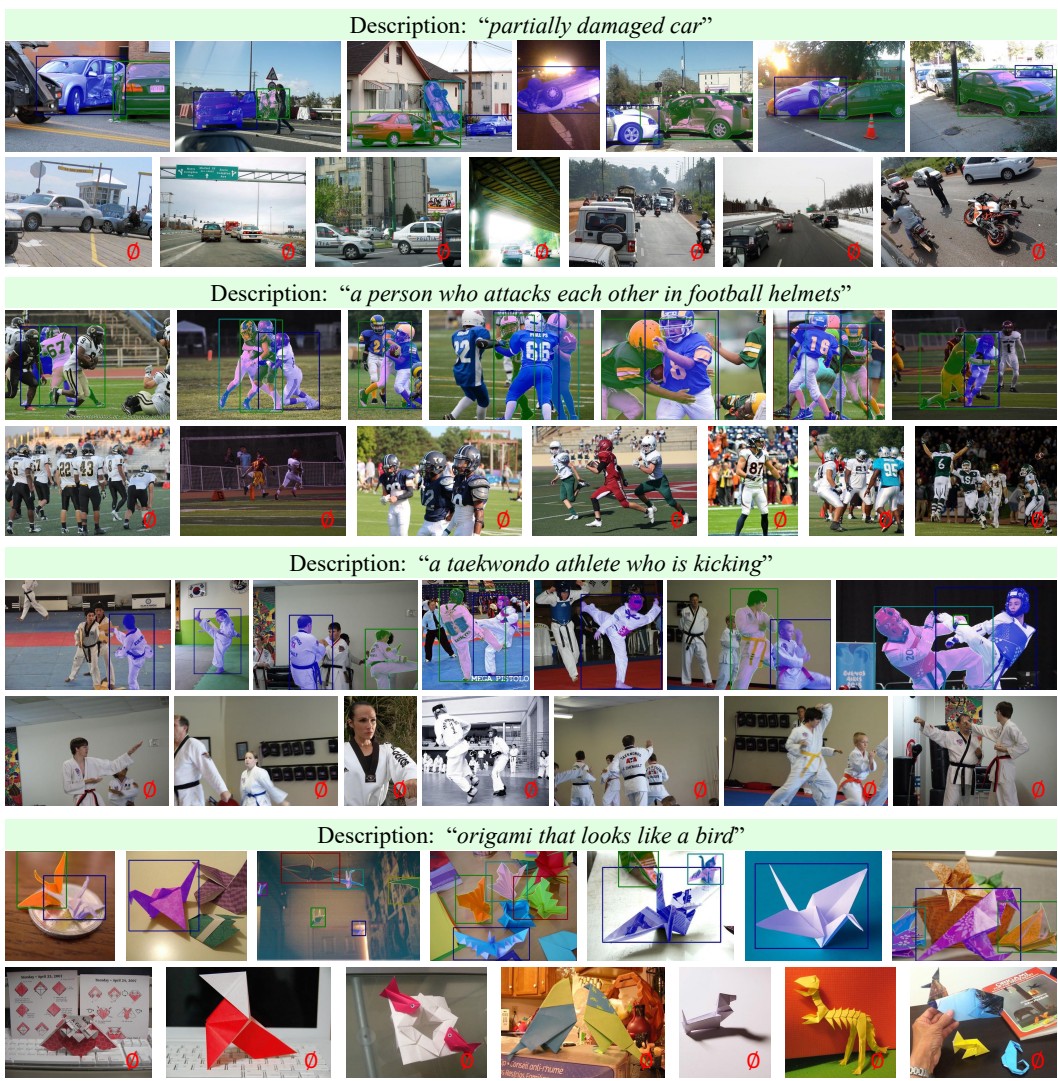

Figure 1: Examples demonstrate that the proposed $D^3$ is fully annotated with positive and negative examples across the entire dataset. The visualizations include four descriptions along with selected positive and negative image samples from the dataset. Each description is accompanied by two rows of image samples: the first row contains positive images, and the second row contains negative images. For positive images, the specific description's bounding boxes and instance masks are visualized. In contrast, for negative images, an empty set symbol $\emptyset$ is displayed in red at the right corner. The visualizations are best observed in color and with zoomed-in view.

absence descriptions and the corresponding presence descriptions differ primarily in the existence of key attributes. For instance, the first presence description emphasizes black/white boards *with* words written, while the first absence description focuses on those *without* words.

It is important to note that in certain cases, some images contain both absence and presence descriptions. For example, in the first example image of the second presence-absence pair, both dogs led by ropes and not led by ropes coexist. Such instances pose significant challenges, as they require the DOD model to comprehend the absence of concepts in a language description and to discern the subtle differences among instances within an image.

**Other characteristics for instance annotations.** Examples in Figs. 1 to 3 all illustrate some additional characteristics of $D^3$:

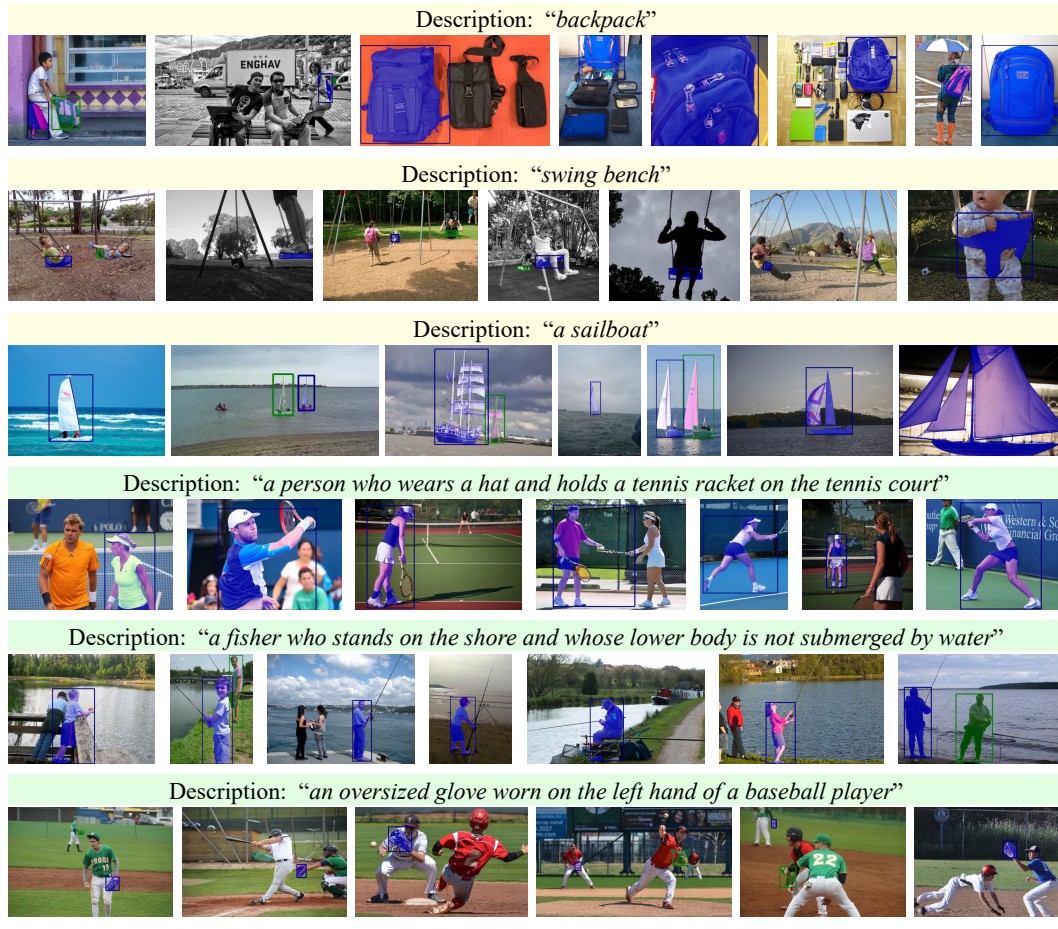

Figure 2: Examples showing the descriptions in D$^3$ are free-form and unrestricted. The descriptions can be short and simple (like the top 3 descriptions, in yellow background) or long and complex (like the bottom 3, in green background). Boxes and instance masks belonging to the specific description are visualized in each image. The visualizations are best observed in color and with zoomed-in view.

(1) Instance-level annotation, where each instance is individually labeled. (2) One description can refer to multiple instances in an image. (3) Each instance is annotated with both bounding boxes and fine-grained masks. As a result, the proposed dataset is not limited to the Described Object Detection setting we investigated in this work. It can also support a similar task, producing instance segmentation masks rather than object detection bounding boxes.

## 1.2 More statistics

The proposed dataset contains a total of 10,578 images, 18,514 boxes (together with corresponding instance masks), and 422 well-designed descriptions. These descriptions comprise 316 presence descriptions and 106 absence descriptions.

Regarding the inter-scenario setting, considering all 422 descriptions, there are 24,282 positive object-text pairs and 7,788,626 negative pairs. When considering only positive descriptions, there are 16,480 positive pairs and 5,833,944 negative pairs.

For the intra-scenario setting (where candidate descriptions for an image only come from the same scenario), there are 20,279 positive pairs and 53,383 negative pairs. For the subset with only positive descriptions, there are 13,917 positive pairs and 41,231 negative pairs.

The average expression length in the dataset is 6.3 words.

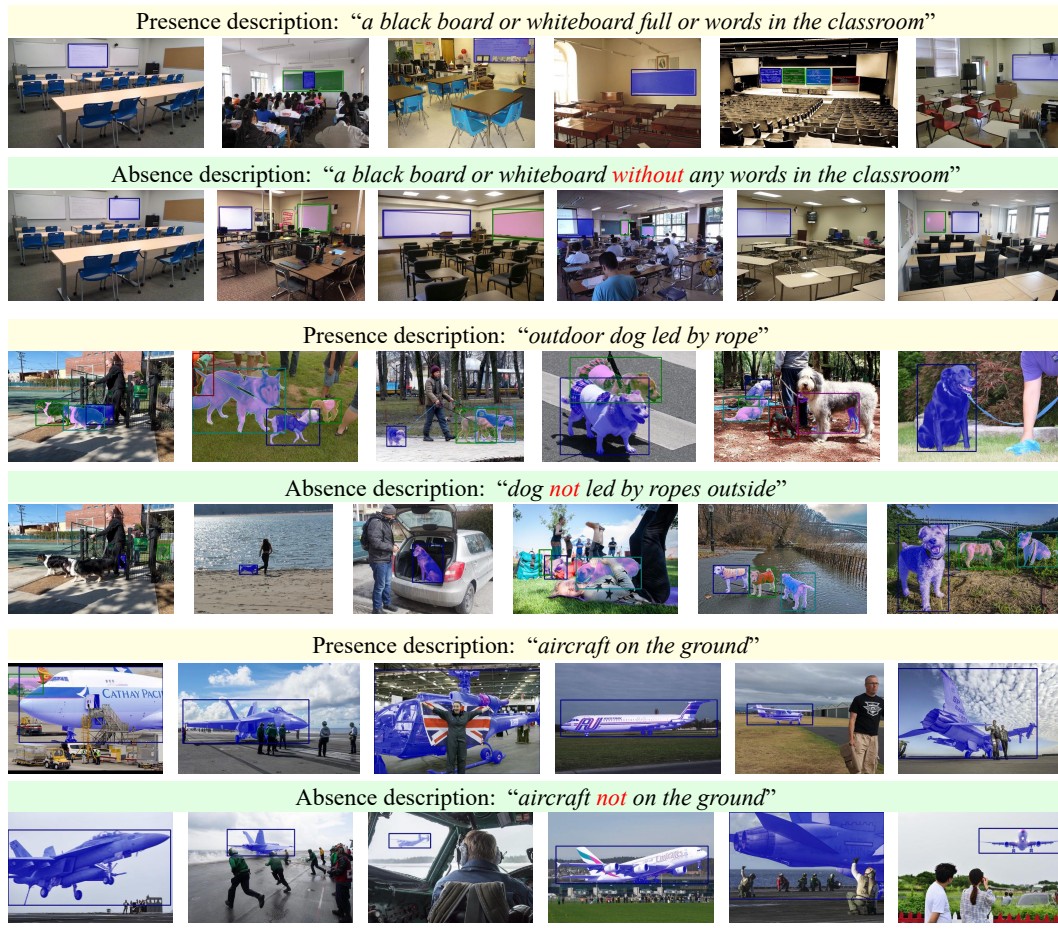

Figure 3: Examples showing the presence and absence descriptions in $D^3$. Six descriptions, containing 3 pairs of contrary presence descriptions (in yellow background) and absence descriptions (in green background), are illustrated alongside their corresponding positive examples. The key words depicting absence expressions are in red. Boxes and instance masks belonging to the specific description are visualized in each image. The visualizations are best observed in color and with zoomed-in view.

In Fig. 4, two additional histograms demonstrate the distribution of the number of positive descriptions and the number of positive instances within a single image in the dataset. This visualization highlights the complexity of the proposed dataset, with frequent occurrences of multiple references and many instances within one image.

**Absence descriptions.** To the best of our knowledge, the proposed $D^3$ benchmark is the first to investigate the capability of models to comprehend the absence of certain features and attributes and distinguish between absence and presence. This unique focus on absence-related comprehension sets it apart from previous benchmarks with description annotation (e.g., datasets like RefCOCO [20, 9] for REC and RES tasks). Notably, RefCOCO contains an extremely small and neglectable number of instances with absence descriptions. In contrast, the $D^3$ dataset comprises 106 absence expressions out of a total of 422 descriptions, approximately 25%, and 7,802 positive annotated instances. This significant inclusion of absence-related expressions contributes to a vital and distinguishing characteristic of our benchmark.

**Category overlapping with previous datasets.** The proposed dataset can be regarded as an OVD benchmark (but with longer references rather than category names), if we take classes and references in previous OVD/REC datasets as *base* classes, and the classes in $D^3$ as *novel*. Categories in $D^3$ have very little overlap with previous datasets. Here we try to quantify the overlap between *base* (OVD datasets like COCO/LVIS and REC datasets like RefCOCO/+/g) and *novel* ($D^3$). For

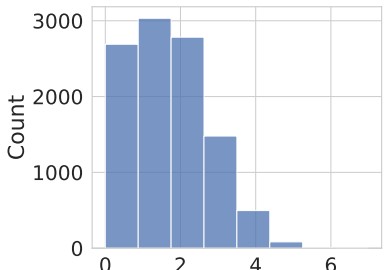
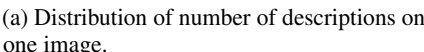

(a) Distribution of number of descriptions on one image.

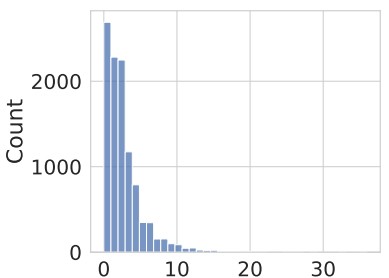

(b) Distribution of number of instances on one image.

Figure 4: Distribution of (a) number of positive descriptions on an image in the dataset, and (b) number of positive instances on an image in the dataset. (a) shows that the majority of images contains multiple positive descriptions in the proposed dataset, while (b) shows that many images contains multiple boxes.

comparison with OVD datasets, we used ChatGPT to generate synonyms from category names in those datasets and then match them against references in $D^3$. The overlapping percentage is 0.4% for COCO and 0.9% for LVIS. For COCO, which have less categories, we also perform manual check and calculation, resulting in 0.7% overlap with $D^3$. For REC datasets, we apply a threshold on the sentence similarity calculated via HuggingFace's `bert-base-cased-finetuned-mrpc` model. The calculated overlaps of $D^3$ with RefCOCO/+/g is 0.0%, 0.2% and 0.7%, separately. Thus, novel classes ($D^3$) overlap <1% with base classes (from OVD & REC datasets).

## 1.3 Annotation process

The data source of $D^3$ is 106 groups from GRD [18], with about 100 images crawled from Flickr and 3~4 designed references for each group. Each group belongs to a different scenario and the overlapping between references from different groups are small (i.e, a reference for one group are not frequent (but possible) to appear in the images from another group). Now we have 10000+ images and 300+ references.

A diagram illustrating the annotation process of $D^3$ is presented in Fig. 5. Here we describe the details of the annotation steps as below:

1. MANUAL Adding absence references: design 1~2 absence references based on the images for each group and add them to the corresponding groups. Now we have 400+ references.

2. AUTOMATIC Selecting possible positive references: for each image, select *all the references* (4~6) from the group it belongs to, and also the other 105 groups (top-$n$ references out of 400+ references, by CLIP [12] similarity between the image and each description). Now for each image, we have $n + 4$~$n + 6$ candidate references and all the other references are filtered out. $n$ is set as 40 initially.

3. MANUAL Verification: randomly choose 5 groups of images, and check if there are any positive references that should not be filtered out. If so, increase $n$ to cover that reference and go back to step 2.

4. MANUAL Human annotation: annotation by trained annotators on all images. The annotation of boxes (and instance masks) are instance-level, dataset-wise complete, and includes absence references.

5. MANUAL Quality check: this includes 3 small steps:

   (a) Discarding some images (unsuitable for annotation, e.g., ambiguity) or categories from the dataset. About 8% samples are discarded.

   (b) Quality check on 100% samples. For each group, if image with error is more than 2%, it is returned for re-annotation. Otherwise the errors are fixed and this group passes this step.

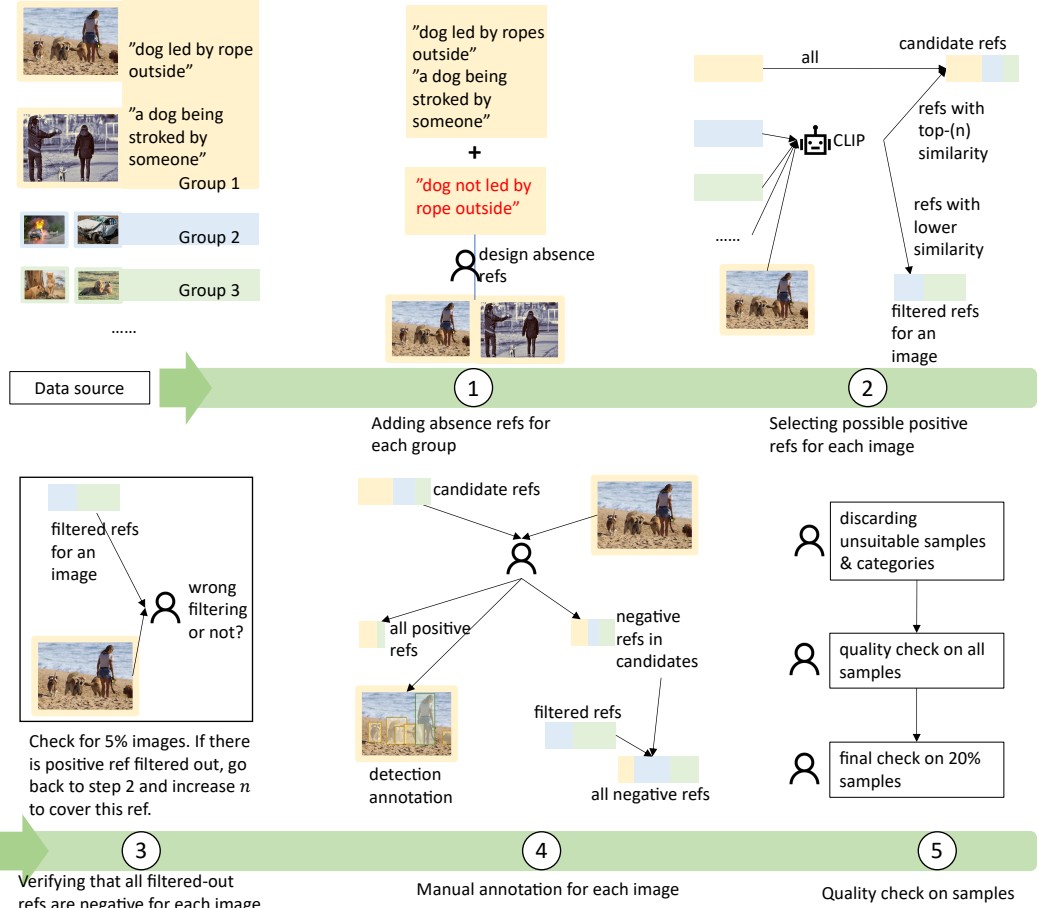

Figure 5: Annotation process of the proposed $D^3$ benchmark.

(c) Final check on 5% samples. For each group, if there are image with error, it is returned, otherwise it is accepted.

**Does utilizing CLIP harms the annotation quality?** As described above, we provide manually annotated certificates for all categories on all images. Such exhaustive and complete annotation is possible by limiting it as an evaluation benchmark (without proposing training set), and utilizing CLIP. But we do not rely on CLIP for deciding a category is positive or negative for an image. It merely provides some initial candidate references, and some designs in the process ensures this: (1) references likely to be positive are bound with the image's group and always kept as candidates, (2) the selection percentage of CLIP is large (>10%) and adjusted based on manual check, (3) the references not selected by CLIP is manually checked by annotators to be negative, (4) the annotators decide a reference is positive or negative, (5) the final annotations are checked by annotators twice. So CLIP only serves as a tool for accelerating the annotation process without deciding the positive/negative, or harming the annotation accuracy.

## 2 Evaluating Existing Baselines

In Section 4.1 of the paper we evaluate several representative and SOTA methods for OVD [10, 17], REC [16] and bi-functional methods [7, 19] on $D^3$ for the DOD task. Here we introduce these methods and describe how we adapt them to DOD and evaluate them on $D^3$. Notably, the images in $D^3$ do not overlap with the training data of these existing baselines and our proposed baseline, so all the comparisons are actually conducted under zero-shot setting, and is relatively fair.

**OFA.** OFA is the SOTA REC method. It is proposed as a general-purpose vision-language model, with ability to performing various tasks like image captioning (IC), VQA, REC, etc. It adopts data from various tasks for pretraining, including MLM, IC, VQA, REC, and OD. Notably, though pretrained on object detection datasets [6, 2], it is not evaluated on these tasks at all. We find that a pretrained OFA-base model merely achieves 9.6 mAP on COCO [6] benchmark, which is too far from modern object detectors. This is also the reason we do not include it as bi-functional models.

OFA can be evaluated on a downstream task either after pretraining or after fine-tuning on the specfic dataset. On REC datasets, it is already strong with only pretraining and achieves SOTA performance after fine-tuning on REC only. As the images in $D^3$ do not overlap with those in REC datasets, we use the pretrained model of OFA rather than the one fine-tuned on REC data, for better generalization ability. The official checkpoints are used as the model to evaluate on $D^3$. Model checkpoints of multiple sizes are available and we use the largest two, namely OFA-base and OFA-large.

For REC task, OFA takes in a pair of one image and one sentence, and predicts a sequence of 4 coordinates, which forms a bounding box. For DOD, we apply a similar inference strategy. For a image and the candidate descriptions (for intra-scenario setting, only a few descriptions in that scenario; for inter-scenario setting, all the descriptions in the dataset), each description and the image form a input image-text pair and predicts a detected instance (bounding box) that will be saved as the result. As OFA predicts token sequences of box coordinates and no classification scores, we use the average of the classification score on the 4 coordinate tokens as the confidence score for each detected instance. No further processing is applied.

**OWL-ViT.** OWL-ViT [10] and CORA [17] are the SOTA OVD methods. OWL-ViT also adopts a pretraining and fine-tuning strategy for training. It is pretrained with image-text contrastive learning, similar to CLIP [12] and then transferred to OVD with simple modification and fine-tuning on standard detection datasets. For evaluation on $D^3$, we use the model fine-tuned on detection datasets without other training. Model checkpoints with ViT-base [1] and ViT-large backbones are available.

For OVD, OWL-ViT takes in some text sequences and one image, and predicts a lot of instances consisting of bounding boxes, class labels as well as classification scores. The text sequences are category names like `giraffe`, `car`, etc. The detected instances with a score less than threshold 0.1 are filtered. For the proposed DOD, we apply a similar inference strategy. The input text is the candidate descriptions, and the output instances are filtered by the same threshold 0.1. No other modifications or post-process are applied.

**CORA.** CORA [17] is a DETR [3] style method that adapts CLIP [12] to OVD. It takes CLIP as the pretrained model and fine-tune the modified framework on detection datasets [6, 2].

The inference of CORA on OVD is performed as a matching between image region features and category name embeddings encoded by CLIP text encoder. For inference on DOD, we adopt the same strategy. We only replace the input images with those from $D^3$ and the category names with the candidate descriptions. Other details follow the settings in CORA for OVD.

**Grounding-DINO.** The bi-functional Grounding-DINO [7] extends a close-set object detector to open-set object detection. It is pretrained on vast object detection [6, 2, 5, 13] and image captioning data [14, 15, 11]. However, this model is not competitive on REC, and a further fine-tuning on REC data [20, 9]is required to achieve a strong performance. Official model checkpoints with Swin-tiny [8] and Swin-base backbones are available.

Grounding-DINO produces a lot of detected instances for one image-text input, and filters some instances with a threshold hyper-parameter. For inference on REC, given an image-reference pair, it merely keeps the one and only instance with the largest score. We follow its inference process on REC task for the proposed DOD. We will dig more into the specific inference strategy and hyper-parameters in the additional experiments in Sec. 4.

**UNINEXT.** UNINEXT [19] stands as another bi-functional method, reformulating a diverse array of tasks, such as object detection, REC, video-based tracking, image and video segmentation tasks, into a unified multi-task framework that excels in instance prediction and retrieval. This innovative approach involves three stages of pre-training without any single-task fine-tuning. In the first stage, training is performed with Object365 [13], followed by the second stage with REC data and COCO, and finally, the third stage with extensive data from video tasks. For evaluation on $D^3$, we utilize the UNINEXT models trained in the second stage, which only utilizes image data and is relatively fair

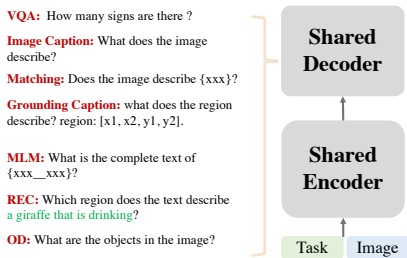

Figure 6: Model structure of OFA [16].

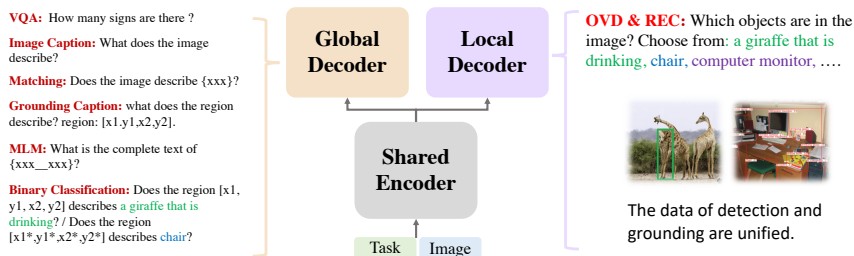

Figure 7: Model structure of the proposed OFA-DOD.

for comparison. Model checkpoints featuring ConvNeXt-large and ViT-huge backbones are available, and these are the ones we employ for evaluation.

For each task it is pretrained on, UNINEXT designs an individual inference strategy. For the DOD task, we adopt an inference strategy similar to REC. To delve deeper into the specific inference strategy and hyper-parameters, we also conduct additional experiments in Sec. 4.

## 3  The Proposed Baseline

As stated in Section 4.2 of our paper, we choose OFA as the foundation for the proposed baseline. Here we provides two figures to show the differences between OFA [16] in Fig. 6 and the proposed OFA-DOD in Fig. 7.

As shown in the two figures, the first modification, granularity decomposition, corresponds to replacing a shared decoder with two parallel decoders, one for global tasks and one for local tasks; the second modification, reconstructed data, refers to the reconstructed OVD & REC data for the local decoder, after which the input can be one or multiple references (or object category names) and they can corresponds to zero, one or multiple targets; the third modification, task decomposition, is depicted by adding a binary classification in the global decoder, which determines if a bounding box and a description is matched.

More details regarding these 3 modifications are stated below:

### 3.1  Granularity decomposition

The aim of this adjustment is to enhance the suitability of the baseline for localization tasks such as OVD, REC, and DOD. The original OFA [16] consists of a multi-modal encoder and a decoder. For each task, whether it involves image-only, text-only, or image-text inputs, an image (which can be omitted) and a text prompt are fed into the multi-modal encoder to predict the output as a text sequence. All task processes are forced to co-exist in one encoder and one decoder.

To achieve this decomposition, we divide the pretraining tasks of OFA into two different granularities: global tasks for language modeling-related tasks like IC, VQA, MLM, etc., and local tasks for region localization-related tasks such as object detection and REC. We add an extra decoder alongside the

original one, which also takes input from the encoder. The two decoders handle global and local tasks independently, thereby avoiding mutual interference.

This improvement effectively resolves conflicts between different tasks and enhances the capability of the model for localization tasks.

## 3.2   Reconstructed data

This improvement is to benefit detection with multiple target instances. For OFA, REC is performed with one image and one text prompt (question prefix concatenated with one description) as input, and a bounding box sequence with 4 coordinate tokens as output. The input sequence has the form:

<p align="center">Which region does the text [REF1] describe? [IMG1],</p>

where [REF1] is a description annotated for the image, and [IMG1] is the image token sequence.

Originally, each input example in REC is a image-text-box pair, where one reference is annotated with one bounding box for one image. We reconstruct the data of REC by 2 steps: First, we group the descriptions belonging to one image, and each reconstructed input example is a combination of one image, $N$ positive descriptions, and $N$ boxes, where $N$ is a integer equal to or larger than 1. Second, for each image, we sample some descriptions from other images as the negative description. With the prepared data, we change the input as:

<p align="center">Which of these options are in the image? Choose from options: [REF1] [REF2] [REF3] ... [IMG1],</p>

where [REF1] [REF2] [REF3] are positive or negative randomly sampled. The output is to predict a series of multiple boxes, each followed by its corresponding descriptions in the input. This results in a unified data format for OD and REC. For OD, the negative descriptions are negative class names. The reformulated data are noisy, as they are not initially prepared for DOD, and a sampled negative description is not necessarily negative due to the image-level annotation completeness of REC. Still, we find such reconstructed data helpful.

## 3.3   Task decomposition

This step aims to enhance the baseline's capability to discern false positives. In addition to training on REC (to locate a region based on a reference), we leverage the multi-task nature of OFA and introduce an additional VQA task. This task involves determining whether a predicted region and a description match with each other and can be viewed as a binary classification problem. The input for this VQA task is:

<p align="center">Does the region [BOX1] describes [REF1]? IMG1,</p>

where [BOX1] is the bounding box coordinate tokens corresponds to the description. For training, the box and the reference are either from a GT text-box pair, or the GT box is shifted (as negative sample), or the box and the reference are from different text-box pairs (as negative sample, too). The output of this task is a text sequence yes for positive samples and no for negative samples. This step is responsible for rejecting possible false positives.

# 4   More experimental results

## 4.1   Additional evaluation results for DOD

**More comparison between baselines.** In Tab. 1 we show a more complete comparison of the evaluated baselines on $D^3$ with different metrics. Results on average recalls are added. In REC datasets like RefCOCO [20, 9], the standard metric is accuracy (which equals to precision and also recall in REC setting). This is not suitable for DOD, which is essentially a detection task. Here we also report the average recall metric in COCO API, but it does not necessarily correspond to the effectiveness of a method for DOD, which requires rejecting negative instances while REC does not.

As shown in Tab. 1, REC methods are bad at recall, possibly because it can only predict one instance for one description, no matter how many instances actually exists in GT. OVD methods are also

Table 1: Comparison of different methods on the proposed dataset for different mAP metrics: intra-secnario mAPs, inter-scenario mAPs, and average recalls. "Bi" denotes bi-functional methods.

| Task | Method | Intra-scenario | | | Inter-scenario | | | Average Recall | | |
|---|---|---|---|---|---|---|---|---|---|---|
| | | *FULL* | *PRES* | *ABS* | *FULL* | *PRES* | *ABS* | *FULL* | *PRES* | *ABS* |
| REC | $OFA_{base}$ | 3.4 | 3.0 | 4.3 | 0.1 | 0.1 | 0.1 | 13.7 | 13.5 | 14.3 |
| | $OFA_{large}$ | 4.2 | 4.1 | 4.6 | 0.1 | 0.1 | 0.1 | 17.1 | 16.7 | 18.4 |
| OVD | $CORA_{R50}$ | 6.2 | 6.7 | 5.0 | 2.0 | 2.2 | 1.3 | 10.0 | 10.5 | 8.7 |
| | $OWL-ViT_{base}$ | 8.6 | 8.5 | 8.8 | 3.2 | 3.7 | **4.7** | 13.5 | 13.7 | 13.1 |
| | $OWL-ViT_{large}$ | 9.6 | 10.7 | 6.4 | 2.5 | 2.9 | 2.1 | 17.5 | 19.4 | 11.8 |
| Bi | $UNINEXT_{large}$ | 17.9 | 18.6 | 15.9 | 2.9 | 3.1 | 2.5 | 40.7 | 42.6 | 34.7 |
| | $UNINEXT_{huge}$ | 20.0 | 20.6 | 18.1 | 3.3 | 3.9 | 1.6 | 45.3 | 46.7 | 41.4 |
| | $G-DINO_{tiny}$ | 19.2 | 18.5 | 21.2 | 2.3 | 2.5 | 2.1 | 47.8 | 48.1 | 46.6 |
| | $G-DINO_{base}$ | 20.7 | 20.1 | **22.5** | 2.7 | 2.4 | 3.5 | 51.1 | 51.8 | 48.9 |
| DOD | $OFA-DOD_{base}$ | **21.6** | **23.7** | 15.4 | **5.7** | **6.9** | 2.3 | 47.4 | 49.5 | 41.2 |

Table 2: Performance of bi-functional methods [7, 19], compared with the proposed baseline, under different score filtering thresholds. The mAP under *FULL* setting and the False Positive Per Category (FPPC) on images with no instance for one category are reported as metrics. For methods filtered with different score thresholds, we highlight the rows when they achieve a FPPC similar to our OFA-DOD.

| Method | Threshold | No-instance FPPC (%) ↓ | *FULL* mAP (%) ↑ |
|---|---|---|---|
| UNINEXT [19] | - | 100.0 | 20.0 |
| | 0.4 | 99.3 | 20.0 |
| | 0.5 | 96.5 | 19.9 |
| | 0.6 | 84.0 | 19.7 |
| | 0.7 | 57.8 | 18.1 |
| | **0.8** | **36.0** | **15.7** |
| | 0.9 | 11.5 | 8.7 |
| Grounding-DINO [7] | - | 100.0 | 20.7 |
| | 0.4 | 80.8 | 20.2 |
| | 0.5 | 60.6 | 18.4 |
| | 0.6 | 45.2 | 16.2 |
| | **0.7** | **34.6** | **13.6** |
| | 0.8 | 23.3 | 9.5 |
| | 0.9 | 8.5 | 3.8 |
| OFA-DOD | - | **35.6** | **21.6** |

bad at this metric though they produce a dozen of output (see Figs. 8 and 9. This may partially explains its low mAP. The bi-functional methods and the DOD one are all strong on this metric. Grounding-DINO, though performs not as good as the proposed OFA-DOD in terms of mAPs, obtains the best recall. This indicates that it tends to produce more detection results.

**Inference of bi-functional methods.** As discussed in Section 5.1 of the main paper, bi-functional methods obtain a 100% No-instance FPPC and fail to reject negative images on $D^3$. This is due to the inference strategy based on REC. It is possible to apply other inference strategy for them.

We verify the effect of inference strategy on these two bi-functional methods [19, 7], with No-instance FPPC and overall *FULL* mAP, and make comparison with the proposed baseline. As shown in Tab. 2, we try to apply a threshold to filter out certain low-score predictions, similar to the post-processing steps in OVD [10]. With this inference strategy, we observe that the increase of score threshold does lower the No-instance FPPC significantly, but at the cost of overall mAP. Therefore, we apply the REC-based inference strategy for these bi-functional methods by default.

Furthermore, we find that when the score threshold is quite high (0.7 for Grounding-DINO and 0.8 for UNINEXT), they reach a FPPC similar to the proposed baseline but with much less overall mAP (15.7 mAP for UNINEXT and 13.6 mAP for Grounding-DINO, while ours 21.6 mAP). Therefore, it might be fair to say that the proposed baseline achieves a better balance between the ability to reject negative images and the overall detection capability.

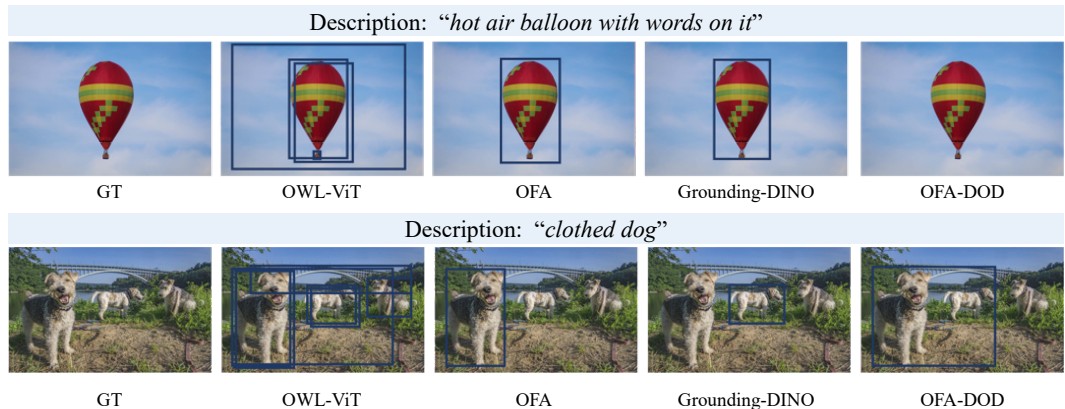

Figure 8: Visualization of detection results from different models on negative images for some descriptions. There is no GT instance on these images for the descriptions. From left to right: GT, predictions from OVD, REC, bi-functional, and DOD methods. Best viewed in color and zoomed in.

## 4.2  Visual comparisons

**Rejecting negative samples.** As shown in Fig. 8, we visualize two descriptions and two images with no corresponding GT instance. An ideal DOD method should refrain from predicting instances on them. OWL-ViT [10], the OVD method, predicts multiple instances on these images, some of which overlap with each other. Such redundant predictions are not suitable for this setting. OFA [16], the REC method, always predicts an instance for one reference, making it highly prone to mistakes in such negative images. Grounding-DINO [7], the bi-functional method, correctly locates the `hot air balloon` and `dog` but fails to capture features related to `with words` and `clothed` in the language description. In the last row, the proposed baseline for DOD successfully rejects one negative image but fails with the other one. This implies that it may perform better on such challenges compared to previous methods, but is still far from being strong.

**Absence or presence descriptions.** In Fig. 9, we present the detection results for two pairs of descriptions, each with one absence description and its exact counterpart presence description. We visualize the ground truth instances and the predicted ones from 4 representative methods.

In the first pair, `a butterfly that` *`doesn't`* `stop on flowers`, the GT exists for the absence description, but not for the corresponding presence counterpart. We observe that previous methods are not sensitive to the distinction between presence and absence, leading to similar results for both descriptions. However, the proposed baseline stands as an exception by correctly predicting the bounding box for the absence description and successfully rejecting the presence one. This could be attributed to the language comprehension ability of OFA, as it is trained on multiple text-related tasks.

In the second pair, `a person in santa claus clothes` *`without`* `bags`, most methods also yield similar results for both descriptions. Although OFA produces noticeably different bounding boxes for two descriptions, the one corresponding to the absence description is overly large, while the one for the presence description results in a negative prediction. Unfortunately, the proposed baseline incorrectly rejects the predictions for this case.