# OpenReview forum: "Described Object Detection: Liberating Object Detection with Flexible Expressions"
_NeurIPS.cc/2023/Conference — NeurIPS 2023 poster_

### Official Review · Reviewer_wkHv · 2023-06-27

**Soundness:** 4 excellent
**Presentation:** 2 fair
**Contribution:** 3 good
**Rating:** 7
**Confidence:** 2

**Summary:**

The paper presents a new multi-modal computer vision task, called Described Object Detection, which is a superset of existing OVD and REC tasks. In particular, the DOD task seeks to create models which can detect multiple instances of something in images, from textual descriptions, which could include describing the absence or presence of something. The paper then goes on to create a new dataset for this task, based on an existing one, called $D^3$. It then shows how existing OVD, REC, Bi-Functional, and proposed a baseline method perform on the data set.

**Strengths:**

The paper is attacking a significant problem, is of high experimental quality, and is novel. The idea of having free-text descriptors for finding things in images is extremely important to the adoption of ML for various important, real-world tasks. In fact, I was surprised to learn that given the importance of this concept, something like DOD had not been proposed before. The paper is also of high quality in that it not only identifies this shortcoming of real-world, open-set detection tasks (i.e. DOD), but also creates a dataset, tests it across a range of existing methods, and proposes a new baseline. Because the paper does propose a new formal problem definition for something that is desired in the real-world performance of ML systems, and therefore realizes how current tasks like REC and OVD fall short of what is actually needed for things like open-set, zero-shot object detection it is a novel look at the problem domain. Finally, it is also worth mentioning that the inclusion of the absence examples is a novel idea, which shows in the poor performance of models on absence instances in the data set.

**Weaknesses:**

The only major weakness of the paper is its clarity. This lack of clarity manifests primarily in two places. First, the description of the creation of the data set needs more detail. Specifically, how was CLIP utilized for adding complete annotations? Were full images given to CLIP with all of the possible annotations and then the most probable selected as an annotation for the image? And, if so, how did you deal with multiple annotations being present in the same images? Additionally, how were the negative image annotations created? Were these done by hand, and, if so, by how many people? Similarly, how many people were involved with creating multiple-instance bounding boxes? The annotation section might benefit from a diagram to better explain this process.

Second, the baseline method is poorly described. It's not clear from the main paper or the semantic similarity how the different proposed components come together into the OFA-DOD model. The paper would greatly benefit from a figure displaying the OFA model and the OFA-DOD model to better understand the architecture of the OFA-DOD model. As it stands now, a reader would have a hard time recreating the OFA-DOD model from the given text.

Finally, there are some minor typo issues that need to be addressed. For example, line 130 page 4 repeats “their” twice, and line 212 on page 8 has the wrong subject-verb agreement (i.e. “fails” instead of “fail”). The paper could use another proofread for typos and grammar.

**Questions:**

Please see the weaknesses section on the questions related to the annotation process and the OFA-DOD model.

**Limitations:**

The paper adequately addresses its limitations and potential negative societal impact.

---

> ### Author Rebuttal · Authors · 2023-08-10
>
> We thank the reviewer for the positive feedbacks.
> **Due to the page limit, please refer to the *[general response (author rebuttal)](https://openreview.net/forum?id=0hwq2vOHT4&noteId=EtVOyLQxeQ)* and the PDF file there for the description and diagram of annotation process of D3 dataset, and the figures showing the model structures of OFA and OFA-DOD.**
> We address the comments below.
>
> ### 1. Clarity on the creation and annotation process of the proposed dataset.
>
> Thanks for the helpful suggestion.
>
> We have added a diagram to show the annotation process, along with text to describe the details of each step, in the ***[general response (author rebuttal)](https://openreview.net/forum?id=0hwq2vOHT4&noteId=EtVOyLQxeQ)*** (diagram in the PDF). We sincerely hope the reviewers will look into this, and believe this diagram and the text will be able to answer the previous questions regarding the clarity on annotation process.
>
> Additionally, we answer the specific questions related to the annotation process as below:
>
> - [Simple explanation of the annotation process] The image are divided into different groups and the description in different group are unlikely to appear in images from other group. For each image, the refs in its group are used. Refs from other groups may also appear, but with a smaller probability, so we use CLIP for selecting a large number of candidates from these refs from other groups. We manually check by statistics that such CLIP filtering usually do not miss positive refs. Then annotators select the positive refs from these refs (rather than all refs in the dataset) and add boxes for each image.
> - [How to use CLIP] CLIP is used merely to filter the references from other groups and decide some negative refs (not all). The refs kept are candidate refs.
> - [How to decide positive and negative for an image, and how to deal with multiple annotations in a image] The annotators select the positive refs (one or more) from candidate refs. Candidate refs can be both the refs from the image's group and the refs filtered and kept by CLIP. They add boxes manually.
> - [How were the negative image annotation created] For each image, the negative refs are (1) those filtered out by CLIP, which the annotators will check and make sure no positive exists, or (2) the candidate refs decided as not positive by the annotators. So, all refs not labeled positive are negative, and they all have manual negative certificates from the annotators.
> - [Human annotation cost] The number of person involved in the annotation process:
>    - Data source and step 1: the authors (3 people, 3 days)
>    - Step 2: automatic, by programs (with CLIP)
>    - Step 3: trained annotators (5 people, 1 week)
>    - Step 4: trained annotators (15 people, 3 weeks)
>    - Step 5: trained annotators (15 people, 2 weeks)
>
> As the annotation process is rather complicated, we recommend to look into the annotation process description in the general response first.
>
> ### 2. Clarity on the proposed OFA-DOD baseline and a figure of model structure.
>
> Thanks for the question. We did not elaborate the details of OFA-DOD due to limited pages of the manuscript and its less important contribution compared with other parts, so we provided a more detailed description on the components differentiating OFA and OFA-DOD in the supplementary material, including the model structure, training setting and inference strategies.
>
> In this response, we provides figures showing the differences between OFA (Fig.2) and OFA-DOD (Fig. 3) in the author rebuttal PDF file. As shown in the figures,
>
> - The first modification, granularity decomposition, corresponds to replacing a shared decoder to two parallel decoders, one for global tasks and one for local tasks. This is likely to alleviate the conflicts between tasks of different granularity and make the model more suitable for localization tasks.
> - The second modification, reconstructed data, refers to the reconstructed OVD & REC data for the local decoder. After the reconstruction, for both task, the input can be one or multiple references (or object category names) and they can corresponds to zero, one or multiple targets.
> - The third modification, task decomposition, is depicted by adding a binary classification in the global branch. This task determines if a bounding box and a description is matched, and is used as a second step to reject negative instances in inference.
>
> ### 3. Some minor typo issues.
>
> Thanks for the suggestions. We just fixed the typos you mentioned together with a few others we found and proofread the whole paper several times. Thank you again for the detailed suggestion.

---

> > ### Comment · Reviewer_wkHv · 2023-08-21
> > **Reply to Rebuttal**
> >
> > After having read the rebuttal to my questions, as well as the replies to the other reviewer's questions, I have no further questions at this point and I will stand by my current evaluation score.

---

### Official Review · Reviewer_YHXi · 2023-07-05

**Soundness:** 3 good
**Presentation:** 3 good
**Contribution:** 3 good
**Rating:** 4
**Confidence:** 5

**Summary:**

In this paper, the authors propose the definition, dataset, evaluation metrics and benchmark results of a new task - Described Object Detection (DOD). DOD is designed to detect objects aligned with full, presence or absence langauge descriptions. The proposed evaluation metrics have three groups based on full, presence or absence descriptions. The authors also propose the Intra-scenario mAP that only detect the categories appear in the image, and Inter-scenario mAP that detecting all the categories. The authors provide the benchmarks results on the proposed task with a new OFA-DOD baseline.

**Strengths:**

- The authors analysize the limitation of previous tasks
- The authors provide comprehensive benchmark results on the proposed new settings

**Weaknesses:**

First, I want to clarify two different tasks: category-level object detection and visual grounding. The category-level tasks like OVD focus on recognizing pre-defined C categories, e.g., "cat," "dog," etc. These categories are commonly mutually exclusive (COCO) or share a parent-child relationship (LVIS). The OVD task will divide the C categories into base and novel two disjoint groups and the models are required to train on base classes and test on novel classes to verify the generalization ability. Instead of detecting C classes, the grounding tasks like REC, aim to align each region to a text phrase. The REC datasets like RefCOCO do not pre-defined a set of categories.
Based on my personal understanding on the OVD and REC tasks, I think this paper has the following weaknesses:

1. The authors argue that DOD is a superset of OVD (#35 and the right part of Figure 1). But I disagree with this argument. The OVD task focuses on evaluating the generalization ability. However, for the DOD task, the proposed dataset evaluation metrics do not consider dividing base/novel groups. Thus, OVD is suitable to evaluate the generalization ability, but DOD is not suitable. If the authors think that the DOD task does not focus on the generalization ability, they should not argue that "superset of OVD".
  - 1.1 For the proposed D3 datasets for DOD, the authors pre-defined 422 categories. What happens to the other expressions that are not pre-defined? The authors may add some novel classes in the test set that are not available during training to simulate this situation. I think the base/novel setting is one advantage of OVD over DOD. If the proposed D3 dataset has already some novel expression in the test set, the authors may consider adding evaluation results on base/novel groups.
  - 1.2 The left part of Figure 1 is the definition of object detection rather than open-vocabulary object detection with the base/novel setting.
  - 1.3 From my perspective DOD is focused on fine-grained language understanding (e.g., the unrestricted language description in #123-127), which is not the main focus of OVD.

2. #188-122: For the REC task, the model is required to align the region with the input language expression and the datasets of the REC task. In my opinion, I don't consider "complete annotation" to be a disadvantage of REC. For instance, REC datasets such as RefCOCO & RefCOCO+ consist of 141,564 language queries, which is significantly larger than the proposed D3 dataset with only 422 language queries. Can we conclude that the vocabulary size of REC datasets is larger than that of DOD? Can we argue that due to the cost of annotation, the "complete annotation" requirement restricts the vocabulary size?

**Questions:**

3. In the proposed DOD task, the category-level definition is vague, and let me feel confused. The authors pre-defined 422 expressions or categories and use these 422 categories for evaluating the Inter-scenario mAP. But are these categories mutually exclusive or overlapped? For example, I guess some short expressions (e.g., 'backpack') may be a parent node of other long expressions (e.g., 'backpack with yellow color'). The authors may consider this point when designing their evaluation metrics, like adopting the positive/negative labels in the OpenImage and LVIS datasets.
- 3.1 Figure 3 (b) and (c) use the "number of instances", does it mean "number of categories"?

4. The motivation for the new DOD task is not clear to me. Can the authors provide straightforward application scenarios or real-world examples that previous tasks are limited while the proposed new DOD task offers broader possibilities?


**Limitations:**

In #341-345, the authors have discussed their limitations regarding potential abuse and carbon emissions.

---

> ### Author Rebuttal · Authors · 2023-08-10
>
> Thanks for the comments. **Due to the page limit, please refer to the *general response (author rebuttal)* for the motivation of DOD task.**
>
> ## 1. Is DOD a superset of OVD or not?
>
> We want to clarify that DOD is a superset of OVD: The D3 dataset is designed solely for evaluation and does not include a training set, so models are trained on OVD/REC datasets and then evaluated on D3. Since the descriptions in D3 has little overlap with existing datasets, the categories in D3 can be regarded as novel categories relative to the training categories (regarded as base) in OVD/REC datasets.
>
> The evaluation of D3 is conducted in a zero-shot/open-vocabulary manner, which is analogous to the evaluation of novel categories in OVD. So DOD also "focuses on the generalization ability." Based on the clarification in the reviewer's comments, DOD does qualify as a superset of OVD.
>
> ### 1.1 Base/novel split in D3.
>
> We can regard all classes in training datasets (OVD, REC) as "base" and all the classes in D3 as "novel".
> As mentioned above, D3 is exclusively designed for evaluation, and its categories has little overlap with the categories in the training datasets (OVD, REC, etc.), given that the 422 reference categories in D3 are specifically designed. Consequently, D3 can be seen as a test set comprising novel categories, which corresponds to the novel split in OVD. Therefore, the results reported on D3 so far can be regarded as zero-shot generalization results on "novel" categories.
>
> We add a table to clarify this:
>
> | task | training set | test set |
> | --- | --- | --- |
> | OVD (e.g. COCO OVD) | base group in OVD dataset (e.g. COCO-base) | novel group in OVD dataset (e.g. COCO-novel) |
> | REC (e.g. RefCOCO) | training split of REC dataset (e.g. RefCOCO train set) | testing split of REC dataset (e.g. RefCOCO test set) |
> | DOD | Existing OVD/REC datasets (e.g. COCO, RefCOCO, etc.) | D3 |
>
> ### 1.2 Fig. 1 (a) defines object detection rather than OVD (with base/novel setting).
>
> The illustration of Fig. 1 was simplified to facilitate presentation of different tasks. In (a), our primary intention was to illustrate that for OVD (1) only short category names are provided without lengthy descriptions, and (2) a category may or may not appear in an image. It does not  emphasize the aspect of zero-shot generalization (given that both OVD and DOD tasks inherently involve zero-shot scenarios).
> Actually Fig. 1 (a) is not limited to either OD or OVD, as we did not specify whether categories like "oven," "dog," "person," etc. in the example, exist in the training dataset or not. The figure does not focus on generalization, does not distinguish between training and testing and thus does not delve into concepts such as base/novel divisions.
>
> ### 1.3 DOD focuses on fine-grained language understanding, which is not the main focus of OVD.
>
> The DOD task includes short or long descriptions, not always fine-grained. It is the D3 dataset rather than the DOD task itself that focus more on fine-grained descriptions (but still covers different granularity).
> In the case of the DOD task, when dealing with longer descriptions, it indeed involves fine-grained language understanding. However, when descriptions are shorter (especially just one or two words), it is more similar to category-based detection. This can be observed in Fig. 2 (b) of the paper and the examples in supp. Fig. 2. When descriptions are shorter, the task aligns with the essence of OVD, hence making DOD a superset of OVD.
>
> For the proposed D3 dataset, its annotations do lean towards relatively longer descriptions, focusing more on fine-grained understanding. This design choice was intended to introduce higher levels of challenge.
>
> ## 2. In REC and DOD, complete annotation limits the scale of references, so not having complete annotation is not a drawback of REC.
>
> In REC datasets, forgoing "complete annotation" did led to more references defined compared to DOD. However, lacking such complete annotation make REC not applicable to many practical detection-related scenarios (as described in the general response), which makes the DOD task with complete annotation useful.
>
> ## 3. Are categories in D3 mutually exclusive or overlapped? Consider this when designing the evaluation metrics.
>
> These categories are not necessarily mutually exclusive; they can overlap in certain instances (hence, the classification in the DOD task is multi-label, not single-label classification), but there's no hierarchical inclusion relationship.
> When designing the dataset's categories, we deliberately avoided including categories with hierarchical relationships (such as "backpack" and "backpack with yellow color") to prevent D3 from becoming too straightforward in terms of difficulty.
> In D3, for an image, categories not positively labeled are manually verified as negative. In other words, our annotations are exhaustive, eliminating the need for partial negative labels to explicitly denote some negative categories in the federated annotation manner of datasets like OpenImage or LVIS. Therefore, naive mAP is suitable for D3 while for OpenImage and LVIS positive/negative labels needs to be considered.
>
> ### 3.1 number of instances" or "number of categories in Figure 3 (b) and (c)?
>
> As shown in the caption of Fig. (3) (b) and (c), they mean "number of instances". More specifically, Fig. 3 (b) means number of positive instances in all images for one description (i.e., one category), which shows that each description has a sufficient number of instances across the dataset. Fig. 3 (c) means number of positive instances in a positive image for one description, which shows a description can have one or multiple (usually 2 to 5) instances in an image.
>
> Note that for "number of categories", we show the distribution of number of descriptions (categories) on one image in supp. Fig 4 (a).
>
> ## 4. Motivation of DOD and real-world application that previous tasks are limited?
> Please see the ***general response***.

---

> > ### Comment · Reviewer_YHXi · 2023-08-18
> >
> > > Is DOD a superset of OVD or not?
> >
> > 1. Results on *base* classes
> >
> > Before reading the rebuttal, I think the proposed method OFA-DOD is trained on the proposed DOD dataset and evaluated on the DOD dataset, so my concern is about missing *novel* classes for generalization. According to the rebuttal, OFA-DOD is trained on OVD/REC datasets and evaluated on the DOD dataset. So my concern now is about missing the results on *base* classes.
> >
> > For the OVD setting, the evaluation is conducted on both the *base* and *novel* classes. However, the authors only report the performance on *novel* classes (e.g., on their proposed new dataset.) But the performance on *base* classes (e.g., on the Open-vocabulary COCO dataset, which is the training set used for DOD) is not provided. So I think missing the evaluation results on *base* classes is a weakness if the authors still argue that the DOD is a superset of OVD.
> >
> > 2.
> >
> > The authors replied that “D3 is exclusively designed for evaluation, and its categories have little overlap with the categories in the training datasets (OVD, REC, etc.), given that the 422 reference categories in D3 are specifically designed.”
> >
> > The ‘little overlap’ is not a rigorous term in an academic paper. How many classes are overlapped? For long-term expression, is there any synonyms in the proposed new dataset? The authors may provide more details.
> >
> > >   Are categories in D3 mutually exclusive or overlapped? Consider this when designing the evaluation metrics.
> >
> > The authors replied that "When designing the dataset's categories, we deliberately avoided including categories with hierarchical relationships (such as "backpack" and "backpack with yellow color") to prevent D3 from becoming too straightforward in terms of difficulty."
> >
> > **I think the hierarchical relationship is common in real-world scenarios**. Many existing datasets have a hierarchical relationship, e.g., the ImageNet / LVIS/ Open-Images datasets. But the authors deliberately avoid the hierarchical relationship, which raises my concerns about the authors arguing that their new setting is a real-world task.
> >
> > > Other feedbacks
> >
> > Some of my questions are not answered:
> >
> > 1. For the proposed D3 datasets for DOD, the authors pre-defined 422 categories. **What happens to the other expressions that are not pre-defined?**
> >
> > 2. REC datasets such as RefCOCO & RefCOCO+ consist of 141,564 language queries, which is significantly larger than the proposed D3 dataset with only 422 language queries. Can we conclude that the vocabulary size of REC datasets is larger than that of DOD? **Can we argue that due to the cost of annotation, the "complete annotation" requirement restricts the vocabulary size?**

---

> > > ### Author Response · Authors · 2023-08-19
> > > **Response to Reviewer YHXi's Feedback**
> > >
> > > Thank you for the responses.
> > >
> > > ### 1. Missing the results on *base* classes if DOD is a superset of OVD.
> > >
> > > - **1**. By definition, OVD aims to generalize beyond *base* classes during training, and detect *novel* classes defined by an open vocabulary at inference. The definition of DOD is the same, except the classes are unrestricted references rather than only short class names. $D^3$ qualifies as a DOD evaluation set as it provides *novel* classes.
> > >
> > > - **2**. **OVD task primarily focuses on evaluating performance on *novel* classes** after training on *base* classes, and evaluation on *base* is mainly an experimental setting to check the capability on seen classes and the upper bounds on unseen. (Some methods, e.g., MEDet and the SOTA CORA, do not provide LVIS *base* performance). The reviewer also mentioned in the original comment that OVD focuses on the generalization on *novel*, and our zero-shot evaluation on $D^3$ also reflects this perspective.
> > >
> > > - **3**. In our evaluation, different baselines are trained on varying tasks and datasets, making it challenging to assess the performance on their *base* classes.
> > >
> > > - **4**. In response to the suggestion, we **reconstruct **$D^3$** for training and eval.** We splitted $D^3$ and obtained a training set (*base*: 259 classes) and two test sets (*base*: 259, *novel*: 126). We conducted training on *base* and evaluated on both *base* and *novel*. The results are presented in the table below.
> > >
> > > | Model | Novel | Base |
> > > | --- | --- | --- |
> > > | OwlViT | 9.7 | 15.2 |
> > > | OFA_base | 4.3 | 11.3 |
> > > | UNINEXT_huge | 18.6 | 23.8 |
> > > | OFA-DOD | 21.6 | 25.1 |
> > >
> > >
> > > ### 2. Quantify the minimal overlap between *base* and *novel* classes for DOD task on $D^3$.
> > >
> > > Thanks for the suggestion regarding rigor of the expression. We've analyzed category overlap between *base* (OVD dataset: COCO/LVIS; REC dataset: RefCOCO/+/g) and *novel* ($D^3$), by:
> > >
> > > - For OVD, we used ChatGPT to generate synonyms from categories, matching them against $D^3$ references. Overlaps: COCO 0.4%, LVIS 0.9%.
> > > - For COCO, which have less categories, we also perform manual check, resulting in 0.7% overlap with $D^3$.
> > > - For REC, we apply a threshold on the sentence similarity calculated via HuggingFace's `bert-base-cased-finetuned-mrpc` model. Overlaps of $D^3$ with RefCOCO/+/g: 0.0%, 0.2%, 0.7%.
> > >
> > > Thus, *novel* classes ($D^3$) overlap <1% with *base* classes (OVD & REC).
> > >
> > > ### 3. Avoiding hierarchical relationship in the dataset may raise concern on if the new setting is real-world task.
> > >
> > > We argue that avoiding such hierarchy does not affect the real-world attributes. The possible references in real world are infinite and we can only annotate some of them.
> > >
> > > Considering our restricted annotation capacity, **we forgo hierarchy to concentrate more on intricate categories and make the descriptions more diverse and rich.** Keeping hierarchical refs like "backpack" and "yellow backpack" seems redundant compared to others, given their similarity. Due to annotator limitations, our categories are limited. Thus, we prioritize informative, diverse refs.
> > >
> > > **Our detection task is multi-label, not single-label, accommodating complex relationships between refs.** We include intricate references like "clothed dog" and "dog not lead by rope outside," with partial overlap and no simple inclusive hierarchy.
> > >
> > > ### 4. What happens to expressions not pre-defined, except the 422 in the dataset?
> > >
> > > Sorry but we didn't quite understand the reviewer's question. If the reviewer's inquiry pertains to whether the $D^3$ dataset supports evaluation beyond the 422: $D^3$ does not support evaluation for categories beyond those defined, as the ground truth annotations are exclusively available for these categories. These categories represent a subset of reality, used to assess method performance, similar to detection/REC datasets. No dataset we know so far can cover all possible references in the world.
> > >
> > > If the reviewer is inquiring about whether the DOD method supports inference for categories beyond the 422 defined: These baselines could theoretically infer from categories outside the 422, but performance might vary significantly, as seen in zero-shot results differences on the 422 categories.
> > >
> > > ### 5. Is the vocabulary of REC datasets larger than $D^3$ for DOD? Did the need for "complete annotation" limit vocabulary due to annotation costs?
> > >
> > > Yes to both. Though our positive and negative sample count surpasses REC's test set, REC's vocabulary remains notably larger than DOD's due to the complete annotation requirement of DOD.
> > >
> > > Enforcing complete annotation, akin to REC's large reference classes, is unfeasible for us due to high costs. Creating our 422-category dataset involved the contributions of a team of 15, spanning nearly 2 months, costing around $11,000.
> > >
> > > We believe a reference-based dataset with complete annotation can be a valuable community resource. It acts as a starting point, potentially inspiring diverse and comprehensive future DOD datasets.

---

> > > > ### Comment · Reviewer_YHXi · 2023-08-20
> > > >
> > > > Thanks to the authors for providing more details like the overlap between the proposed dataset with previous OVD/Det/REC datasets, which I believe will make a more solid manuscript.
> > > >
> > > > > Missing the results on base classes if DOD is a superset of OVD.
> > > >
> > > > For the OVD dataset, achieving strong results in new classes without compromising performance in base classes is essential. I think it's important to present results for both the base and new splits, as clearly outlined in the OVD setting (for instance, OVR-CNN/ViLD). I appreciate the authors providing more results like reconstructing the proposed dataset for training and eval.
> > > >
> > > > > About the hierarchical relationships
> > > >
> > > > I think the COCO-style mAP evaluation is not suitable for real-world settings. COCO is defined as 80 classes (human-selected and relatively small) and avoids hierarchical relationships.
> > > >
> > > > In recent years, large-scale real-world datasets such as LVIS (with 1,203 classes) and OpenImages (with 500 classes in the v4/v5 version) have demonstrated that as the number of categories increases, challenges related to hierarchy and synonymy arise. This is why both LVIS and OpenImages have introduced new evaluation metrics, such as federated evaluations with positive/negative labels. Such federated evaluations will be also beneficial to avoid the high labeling cost.
> > > >
> > > > In fact, previous datasets such as LVIS and OpenImages are also multi-label. It's natural to use federated evaluation for assessing multi-label datasets. However, the COCO-style mAP evaluation doesn't support multi-label evaluation.
> > > >
> > > > The dataset proposed, which uses a COCO-style mAP evaluation, implies two main constraints: (1) the number of categories must be limited, and (2) the categories must be predefined or selected by humans. This is why I believe such an evaluation pipeline doesn't reflect a real-world task setting. By contrast, the federated evaluation settings in the style of LVIS/OpenImages accommodate a greater number of categories with more complex relationships.

---

> > > > > ### Author Response · Authors · 2023-08-20
> > > > > **Response to Reviewer YHXi's Official Comment**
> > > > >
> > > > > Thanks for reviewing our responses.
> > > > >
> > > > > > 1. About the results on *base* classes
> > > > >
> > > > > Thanks for the feedback. We are happy that the reviewer recognized the results we provided on both *base* and *novel*. We will add the discussion and analysis above in the manuscript. Please let us know if there is any clarification needed.
> > > > >
> > > > > > 2. About the hierarchy relationships
> > > > >
> > > > > We summarize the reviewer's opinion in this part as: the reviewer thought the complete annotation and the corresponding COCO-style mAP metric for the dataset is not realistic, and encourages using federated annotation and evaluation, similar to LVIS, which may brings the benefits of more categories, with hierarchy or synonym relationships and not pre-defined.
> > > > >
> > > > > We response to this opinition with the following points:
> > > > >
> > > > > **(1)** We agree that federated annotation & evaluation brings the benefit of more category and more relationships, **but not the benefit of no pre-defined categories**. For expression-based dataset, we still need to pre-define and human-select categories even with federated annotation and evaluation. This is because expressions, unlike object category names, are infinite on a set of images, due to the versality of language, so **annotators always need to select manually some expressions from the infinite possible ones.**
> > > > >
> > > > > If we simply annotate objects in images with expressions human-selected but not pre-defined (i.e., whatever the annotator comes up with), then we will get a dataset with a lot of categories but one or few instances for each category, similar to a REC dataset, not a DOD dataset. Thus, **a expression-based detection dataset like **$D^3$** requires us to pre-define and human-select expression categories.**
> > > > >
> > > > > **(2)** We argue that **federated annotation & eval or complete annotation & COCO-style eval have different advantages and the choice depends on the situation**. Complete annotation & COCO-style eval brings more positive and negative samples annotated and evaluated (from all the images in the dataset) for each category, while federated annotation & eval brings larger number of categories but less samples (from only a subset) for each category.
> > > > >
> > > > > **(3)** **For DOD, currently we prefer complete annotation & eval over federated annotation & eval, because we need more negative samples with explicit negative certificates** to verify a model's ability to distinguish and reject negative instances, as described in the motivation in the [general response](https://openreview.net/forum?id=0hwq2vOHT4&noteId=EtVOyLQxeQ). A dataset with federated annotation & eval is nice for the advantages, but not the original target of this work.
> > > > >
> > > > > **(4)** Our dataset focuses on utilizing completely annotated samples to validate the capabilities to detect (discover, locate and reject) instances with flexible expressions. Currently, there are other potential aspects not taken into account, such as the influence of expression relationships. We will continue to investigate them. It is conceivable that in the future, we may witness the emergence of datasets based on $D^3$ featuring federated labeling and encompassing a broader range of categories. Our work serves as a kick-off point for DOD.
> > > > >
> > > > > We would like to clarify there that the contribution majorly involves a dataset for detection with flexible expressions, accompanied with task setting, comprehensive experimental analysis and a new baseline. We earnestly hope that the esteemed reviewer takes into account the potential contribution of this work.
> > > > >
> > > > > **(5)** One more thing to note: the reviewer mention that _COCO-style mAP evaluation doesn't support multi-label evaluation_. As far as we know, when the annotation is not complete (with federated annotation, like LVIS/OpenImage), it does not support multi-label eval, but **with complete annotation, COCO-style mAP is feasible**. Examples of multi-label evaluation using COCO-style mAP includes multi-label image classification on COCO and Human-Object Interaction Detection on HICO-Det, to name a few. Our annotation is complete so COCO-style mAP is plausible as the evaluation metric.

---

> > > > > > ### Comment · Reviewer_YHXi · 2023-08-20
> > > > > >
> > > > > > My argument is that the COCO-style mAP evaluation is not suitable for a real-world setting. That’s because COCO-style mAP evaluation, like this paper, needs disjoint labels, and the number of classes cannot be very large.
> > > > > >
> > > > > > > (1) We agree that federated annotation & evaluation brings the benefit of more category and more relationships, but not the benefit of no pre-defined categories. For expression-based dataset, we still need to pre-define and human-select categories even with federated annotation and evaluation. This is because expressions, unlike object category names, are infinite on a set of images, due to the versality of language, so annotators always need to select manually some expressions from the infinite possible ones. If we simply annotate objects in images with expressions human-selected but not pre-defined (i.e., whatever the annotator comes up with), then we will get a dataset with a lot of categories but one or few instances for each category, similar to a REC dataset, not a DOD dataset. Thus, a expression-based detection dataset like $D^3$ requires us to pre-define and human-select expression categories.
> > > > > >
> > > > > > The author may misunderstand my concern. My concern is about pre-defined ‘**disjoint**’ classes. In the real-world scenario, as the number of classes increases, other types of pairwise relationships such as partially overlapping or parent-child will occur. Previous federated evaluations like LVIS/OpenImages support such hierarchy label structures. But this paper uses manual selection to avoid complicated relationships, so I don’t think it’s a real-world setting.
> > > > > >
> > > > > > The authors mention that “annotators always need to select manually some expressions from the infinite possible ones”. In fact, this is just one advantage of the federated setting to reduced workload. Federated datasets do not need exhaustive annotations.
> > > > > >
> > > > > > > (2) We argue that federated annotation & eval or complete annotation & COCO-style eval have different advantages and the choice depends on the situation. Complete annotation & COCO-style eval brings more positive and negative samples annotated and evaluated (from all the images in the dataset) for each category, while federated annotation & eval brings larger number of categories but less samples (from only a subset) for each category.
> > > > > >
> > > > > > I think the latter (larger number of categories but fewer samples) is more like a real-world setting. In fact, as the number of categories increases, some of the frequent classes will have sufficient samples, while some of the rare classes will have insufficient samples, like the LVIS/OpenImages dataset. Previous open-vocabulary object detection tasks also use LVIS as their benchmark for real-world consideration.
> > > > > >
> > > > > > > (3) For DOD, currently we prefer complete annotation & eval over federated annotation & eval, because we need more negative samples with explicit negative certificates to verify a model's ability to distinguish and reject negative instances, as described in the motivation in the general response. A dataset with federated annotation & eval is nice for the advantages, but not the original target of this work.
> > > > > >
> > > > > > The federated evaluation also measures the model’s ability to reject negative instances. You may refer to section 2.2 in the LVIS paper.
> > > > > >
> > > > > > > (4) Our dataset focuses on utilizing completely annotated samples to validate the capabilities to detect (discover, locate and reject) instances with flexible expressions. Currently, there are other potential aspects not taken into account, such as the influence of expression relationships. We will continue to investigate them.
> > > > > >
> > > > > > My concern is that the current evaluation pipeline is somehow not real-world with many restrictions. Glad to see that the authors will consider investigating more complex relationships. I think the federated evaluation is a good choice, which has been demonstrated in previous LVIS/OpenImages papers.
> > > > > >
> > > > > > > (5) One more thing to note: the reviewer mention that COCO-style mAP evaluation doesn't support multi-label evaluation. As far as we know, when the annotation is not complete (with federated annotation, like LVIS/OpenImage), it does not support multi-label eval, but with complete annotation, COCO-style mAP is OK. Examples of multi-label evaluation using COCO-style mAP includes multi-label image classification on COCO and Human-Object Interaction Detection on HICO-Det, to name a few. Our annotation is complete so COCO-style mAP is plausible as the evaluation metric.
> > > > > >
> > > > > > First, I want to argue that federated evaluations like LVIS/OpenImage support multi-label evaluation. You may refer to Figure 2 and Section 2.2 of the LVIS paper.
> > > > > >
> > > > > > Some of the datasets build on MS-COCO support multi-label classification, but they use the federated evaluation setting. For example, the mentioned case, HICO-Det, is also a federated evaluation setting with positive/negative groups.

---

> > > > > > > ### Author Response · Authors · 2023-08-21
> > > > > > > **Response to Reviewer YHXi's Recent Comment**
> > > > > > >
> > > > > > > Thanks for the detailed responses. We greatly respect and appreciate these.
> > > > > > >
> > > > > > > > COCO-style mAP eval is not suitable for real-world setting, because COCO-style mAP, like this paper, needs disjoint labels, and class number cannot be very large.
> > > > > > >
> > > > > > > For COCO-style mAP, we agree that *class number cannot be very large* for annotation cost. As $D^3$ is for eval, we do not aim to develop a dataset with thousands or more classes in this work, and 422 is a balance between dataset scale and cost. We prefer complete annotation with more positive and negative samples as this is the first for DOD.
> > > > > > >
> > > > > > > We want to argue against *COCO-style mAP eval needs disjoint label*: since we use multi-label classification setting, classes with overlap are OK as long as completely labeled like $D^3$ (and many others, like HICO-Det, which has non-disjoint labels but adopts COCO-style mAP and complete annotation). Our dataset provides classes like "clothed dog" and "dog not led by rope outside", which are partial-overlap & non-disjoint but OK for eval, as they are completely labeled.
> > > > > > >
> > > > > > > > ... My concern is about pre-defined ‘disjoint’ classes. In real-world scenario, as class number increases, other pairwise relationships such as partially overlapping or parent-child will occur. Federated evals like LVIS support such hierarchy label structures. But this paper uses manual selection to avoid complicated relationships, so I don’t think it’s real-world setting.
> > > > > > >
> > > > > > > About "disjoint": as in the paragraph above, though we avoided hierarchy (simply to make the classes richer and more diverse), there are other non-disjoint labels like those with partial overlapping.
> > > > > > >
> > > > > > > About "pre-defined" "manual selection": As in the last response, manual selection can be avoided for object name-based detection like LVIS as object names in a dataset are limited to several thousands. But for expressions which can be infinite, there are always manual selection needed.
> > > > > > >
> > > > > > > > The authors mention that “annotators always need to select manually some expressions from the infinite possible ones”. In fact, this is just one advantage of federated setting to reduce workload. Federated datasets don't need exhaustive annotations.
> > > > > > >
> > > > > > > The reviewer seems to agree "annotators always need to select manually some expressions from the infinite possible ones", which explains manual selection is necessary for expression-based dataset like ours. We agree that federated datasets don't need exhaustive annotation.
> > > > > > >
> > > > > > > > I think larger class number but fewer samples is more like real-world setting. As class number increases, some frequent classes will have sufficient samples, while some rare classes will have insufficient ones, like LVIS. ...
> > > > > > >
> > > > > > > We agree that federated annotation is good for larger class number. We will consider using federated annotation if we are to build a dataset with larger class number based on $D^3$. Currently it serves as an eval benchmark and a kick-off for DOD. Thanks for pointing this direction.
> > > > > > >
> > > > > > > > The federated eval also measures the model’s ability to reject negative. ...
> > > > > > >
> > > > > > > We did not suggest that federated eval doesn't measure the model's ability to reject negative. We were claiming complete annotation and eval, compared with federated, provides **more** positive and negative samples and focuses **more** on evaluating that ability. As the first for DOD, we are not able to cover all aspects of an ideal dataset, and focus less on large class number. A federated dataset may be a future work.
> > > > > > >
> > > > > > > > My concern is that current eval pipeline is not real-world with many restrictions. Glad to see that the authors will consider investigating more complex relationships. ...
> > > > > > >
> > > > > > > Thanks for the comments. All your suggestions are helpful to us.
> > > > > > >
> > > > > > > > First, I want to argue that federated eval like LVIS support multi-label eval. ... Some datasets build on MS-COCO support multi-label, but they use federated eval. For example, the mentioned HICO-Det is also federated eval setting with pos/neg groups.
> > > > > > >
> > > > > > > We did not suggest federated eval doesn't support multi-label. What we tried to clarify is **COCO-style mAP is OK for multi-label setting**, for this was queried by the reviewer previously.
> > > > > > >
> > > > > > > Both HICO-Det for HOI and COCO for multi-label image classification adopts complete annotation & COCO-style mAP, rather than federated eval. If the reviewer has questions regarding their eval, references can be found in HICO-Det paper (WACV18) like "For each HOI category, we evaluate detection on the full test set" and "we evaluate HOI detection using mAP", and in the `compute_map` functions of the official implementation of various methods, like QPIC (CVPR21) and PPDM (CVPR20); the mAP for COCO multi-label image classification are available in the `mAP` class in the implementation of ASL (ICCV21). We mentioned these to prove COCO-style mAP is feasible for multi-label.
> > > > > > >
> > > > > > > These discussion are very helpful to this work and we would like to express our gratitude to the reviewer for their detailed comments.

---

### Official Review · Reviewer_EwWL · 2023-07-06

**Soundness:** 3 good
**Presentation:** 3 good
**Contribution:** 2 fair
**Rating:** 5
**Confidence:** 4

**Summary:**

This paper introduces a new Described Object Detection (DOD) task, which extends the existing Open-Vocabulary Object Detection (OVD) and Referring Expression Comprehension (REC) tasks into a more general paradigm. For this new task, the authors build a Description Detection Dataset (D3), and find the troublemakers that currently hinder current REC, OVD, and bi-functional methods. They further propose a baseline that outperforms existing methods on the DOD task.

**Strengths:**

1.	Focusing on language-driven object detection, the authors propose a new DOD task, which they argue is more practical and presents more challenges than the current OVD and REC tasks. They also introduce a new dataset D3 for this task.
2.	On the DOD task, the authors thoroughly investigate the challenges faced by current REC, OVD, and bi-functional approaches, and they put forth a baseline method with state-of-the-art performance on this task.


**Weaknesses:**

1.	While the authors introduce this new DOD task, instead of constructing a new training dataset or setting, they focus on analyzing the performance of models trained on the old OVD or REC tasks for this new task. This may not fairly represent the performance of existing methods for the new task.
2.	As the DOD task can be seen as a more generalized version of the OVD task, where category names are extended to language descriptions, the experimental results and conclusions for OVD methods highly depend on their training data and settings. It would be beneficial to assess OVD baselines trained following the style and setting of the DOD task.
3.	The compatibility of the DOD task with the existing OVD and REC tasks is not studied. Given that the DOD task is essentially a superset of the OVD and REC tasks, it would be beneficial to evaluate the baseline OFA-DOD in comparison with other methods on the OVD and REC tasks.
4.	The existing methods and the proposed baseline are evaluated and compared under a zero-shot setting. How would the performance vary when the models are fine-tuned on the D3 dataset?


**Questions:**

Please refer to the weakness part.

**Limitations:**

The authors have discussed the limitations and broader impacts.

---

> ### Author Rebuttal · Authors · 2023-08-10
>
> ### 1. Analysis of existing methods on DOD may be not fair because this work does not introduce new training dataset.
>
> Our approach focuses on evaluating DOD using OVD/REC-trained models and providing insights for transitioning to DOD, emphasizing differences in training tasks and formats.
>
> Currently, the evaluation is zero-shot for existing baselines. Ideally, we would also provide a dedicated DOD training set. However, due to the substantial annotation costs, we have not included a separate training set. Instead, we offer a dedicated test set with accurate annotations as benchmark.
>
> Furthermore, existing OVD/REC data can be leveraged to train DOD models, given the similarity in task formats between OVD/REC and DOD. Taking into account the huge annotation costs for a DOD training set, creating a new training set would provide limited value, and is not necessary or cost-effective.
>
> Therefore, this work focuses on:
>
> - Comparing generalization performance: We analyze the performance of existing methods on D3, aiming to uncover the capabilities of different models across various types and tasks for DOD. The primary goal is to reveal distinctions between training tasks and formats (OVD/REC/bi-functional), rather than a direct comparison of superiority. Thus, we allow for variations in architecture and data among these models.
> - Providing insights and guidelines: We offer insights and guidelines for transitioning from existing OVD/REC to DOD, and training DOD models using available data. These insights are not limited to a specific baseline. OFA-DOD is to demonstrate that simple adjustments can enhance the effectiveness of an REC model originally unsuitable for DOD.
>
> ### 2. Assess OVD baselines trained following DOD settings rather than original setting.
>
> We modified and trained OVD methods with adjustments similar to those applied in OFA-DOD (i.e., the reconstructed data step, which we reconstructs REC to DOD format). The results are shown in the table below, and the methods trained under DOD setting are denoted as OWLViT-DOD and CORA-DOD. Notably, the performance of these models on D3 surpasses original baselines (OWLViT and CORA). This shows the proposed modification over existing methods like OFA are transferrable to other existing OVD/REC methods.
>
> | Model | FULL | PRES | ABS |
> | --- | --- | --- | --- |
> | OWLViT | 9.6 | 10.7 | 6.4 |
> | CORA | 6.2 | 6.7 | 5.0 |
> | OFA | 3.4 | 3.0 | 4.3 |
> | OWLViT-DOD | 12.1 | 12.8 | 10.1 |
> | CORA-DOD | 7.9 | 8.2 | 7.1 |
> | OFA-DOD | 21.6 | 23.7 | 15.4 |
>
> ### 3. Evaluate OFA-DOD on OVD and REC in comparison with other methods (to study the compatibility of DOD with OVD and REC).
>
> We evaluate OFA-DOD on OVD/REC datasets. The results indicate **substantial improvements over OFA** for both OVD and REC. This shows the improvements of OFA-DOD over OFA make it better for DOD/OVD/REC, and when a model is improved to be more suitable for DOD, it also exhibit corresponding performance gains on REC and OVD. This implies that the DOD task is compatible with REC and OVD.
>
> Compared with SOTAs on REC (without fine-tuning), OFA-DOD outperforms SOTA like G-DINO. Compared with SOTAs on OVD (fine-tuning on base classes), OFA-DOD is not good as CORA, but outperforms Detic on novel classes, showing good generalization ability. We argue the reason why OFA-DOD does not obtain SOTA on OVD is that the original OFA is not suitable for detection tasks, incapable of rejecting negative instances, lacking compatibility with multi-target outputs and yielding poor results. Although OFA-DOD has augmented its detection ability and improved its performance on OVD by more than 20 mAP, it is still far from perfect for OVD and DOD. This is no surprise as it is only a baseline for future research.
>
> REC results:
>
> | Benchmark | refcoco |  |  | refcoco+ |  |  | refcocog |  |
> | --- | --- | --- | --- | --- | --- | --- | --- | --- |
> | Split | val | testA | testB | val | testA | testB | val-u | test-u |
> | OFA-base | 61.00 | 62.24 | 58.59 | 45.02 | 48.36 | 38.70 | 46.96 | 46.96 |
> | OFA-DOD-base | 75.92 | 78.74 | 72.16 | 64.98 | 71.32 | 59.25 | 71.52 | 71.76 |
> | G-DINO-L | 73.98 | 74.88 | 59.29 | 66.81 | 69.91 | 56.09 | 71.06 | 72.07 |
>
> OVD results:
>
> | Benchmark | COCO-OVD |  |  |
> | --- | --- | --- | --- |
> | Split | novel | all | base |
> | OFA | 3.2 | 7.4 | 8.9 |
> | OFA-DOD | 28.4 | 30.1 | 30.7 |
> | Detic | 27.8 | 45.0 | 47.1 |
> | CORA_R50 | 35.1 | 35.4 | 35.5 |
>
> ### 4. How would the performance vary when the models are fine-tuned on D3 rather than zero-shot eval?
>
> This work does not propose a new training set (due to the reasons in Q1, including huge annotation costs and availability of existing detection & REC data), making fine-tuning unfeasible. However, to address the question from the reviewer, we partitioned the D3 dataset into training (80 groups, 238 presence and 80 absence refs) and testing (26 groups, 78 presence and 26 absence refs) subsets and fine-tune various baseline methods on the training subset, subsequently evaluating the performance on the testing subset.
>
> We show the results of methods without or with (*) fine-tuning on the training subset and evaluated on the testing subset (not the complete D3). As the results shows, OVD/REC/bi-functional/DOD methods are improved by certain margins (2 to 5 mAP). Note that D3 remains as an evaluation benchmark, not providing a training set. The models are expected to be trained on datasets such as OVD/REC and then tested on the D3 dataset. The introduced training/testing split here is solely for validation purposes.
>
> | Task | Model | FULL | PRES | ABS |
> | --- | --- | --- | --- | --- |
> | OVD | OWLViT | 9.7 | 10.5 | 7.1 |
> |  | OWLViT* | 13.0 | 13.8 | 10.7 |
> |  | CORA | 5.7 | 5.8 | 5.4 |
> |  | CORA* | 8.6 | 9.1 | 7.2 |
> | REC | OFA_base | 4.0 | 3.9 | 4.1 |
> |  | OFA_base* | 8.2 | 8.1 | 8.4 |
> | bi-func | UNINEXT_huge | 20.2 | 21.7 | 15.6 |
> |  | UNINEXT_huge* | 22.3 | 22.8 | 20.9 |
> | DOD | OFA-DOD | 21.4 | 22.6 | 17.7 |
> |  | OFA-DOD* | 24.2 | 25.8 | 19.5 |

---

> > ### Comment · Reviewer_EwWL · 2023-08-21
> >
> > Thank the authors for their detailed response. I appreciate their efforts in addressing my concerns. However, I still believe that having a proper training dataset is more critical and necessary for this DOD task and could greatly enhance the impact of this work. Using existing OVD/REC data to train DOD models may not fully exploit the true potential of current methods or models. Therefore, I would like to maintain my original rating.

---

> > > ### Comment · Reviewer_Q5JG · 2023-08-21
> > >
> > > I’ve been following this discussion and would like to share my thoughts - given that the paper has quite clearly mentioned that it is purely an evaluation benchmark (with 100k images no less), I don’t think it’s fair to evaluate it on the basis of not having a training set. There are plenty of papers (such as Winoground, SVO-Probes, etc) that have been proposed to uncover the limitations of current VL models and have been quite valuable to the community. As you mention « Using existing OVD/REC data to train DOD models may not fully exploit the true potential of current methods or models. » , I believe this is exactly the thing this benchmark aims to measure, ie the fact that there is no good model that can do both REC and OVD. Overall, I think it’s a good addition to the set of available evaluation benchmarks.

---

### Official Review · Reviewer_SdJt · 2023-07-07

**Soundness:** 3 good
**Presentation:** 3 good
**Contribution:** 3 good
**Rating:** 5
**Confidence:** 5

**Summary:**

Brief Summary: The paper presents a new task Described Object Detection which extends open vocabulary object detection (OVD) to use phrases. This, in turn, extends referring expression (REC) to include objects not seen in the training data. To this end, the authors introduce a new dataset called D3 building on existing GRD dataset [44].

The key idea behind creating D3 are to have complete annotation (i.e. each referring expression has a bounding box in each image if present), have natural language (extends OVD), include absence expression (i.e. objects NOT having a particular feature / attribute such as blackboard with no signs), and one expression referring to more than one instance.

The authors experiment on the provided D3 dataset and provide a detailed benchmark with multiple baselines ranging from REC methods like OFA, OVD methods like OWL-ViT and bifunctional method like Grounding-DINO. The authors further propose a new baseline OFA-DOD which changes some pre-training schemes such as including additional localization tasks and find that it outperforms competitive baselines.


**Strengths:**

Pros:

1. Dataset contribution is always welcome. It is clear the authors have put thought into the dataset construction. In particular, absence expression are quite interesting.

2. A number of baselines are considered and the takeaways are quite interesting that REC methods fail at this task due to being unable to effectively choose more than one boxes. While OVD methods  perform better, there is a very large gap compared to bifunctional methods like grounding-dino. The proposed baseline OFA-DOD makes sense and it is good to see that it outperforms other baselines.

3. Visualization of the dataset as well as results in suppl. are very useful. Ablative studies on the baseline such as effect of training data (Table 5b) are interesting.

**Weaknesses:**

Cons:

1. The authors should compare their work with zero-shot grounding [Ref1] which also extends REC to new objects. The obvious difference is that DOD can have more than one instances but a clear distinction would be helpful.

2. The main idea behind DOD is to encompass both OVD and REC. The authors need to motivate this setting more. In my opinion, having the two cases separate can be much more revealing than trying to combine the two. OVD is strictly object detection (with phrases) while REC explicitly requires disambiguation between different objects. For instance, "oversized glove on left-hand" in suppl fig 2 (last row), is simply object detection of glove and doesn't require reasoning whether it is "oversized" or on "left-hand". To me a more natural setting is to separate the two. I would like to know about the author's motivation for the task.

3. It is unclear in the text but in OFA-DOD which OFA is chosen, base or large? To have a fair comparison with Grounding-DINO it should be base but it is not clear (in main text as well as in suppl).

4. In suppl. Table 2, it seems Grounding-DINO outperforms OFA-DOD by considerable margin on Average Recall. Why is this the case? For practical use, wouldn't one prefer using Grounding-DINO?


[Ref1]: Sadhu, Arka, Kan Chen, and Ram Nevatia. "Zero-shot grounding of objects from natural language queries." In Proceedings of the IEEE/CVF International Conference on Computer Vision, pp. 4694-4703. 2019.

**Questions:**

See weakness section.



**Limitations:**

Yes, the authors have discussed some limitations but more could be included such as:

1. The evaluation is heavily dependent on the choice of 412 phrases and the dataset used for GRD.

---

> ### Author Rebuttal · Authors · 2023-08-10
>
> We thank the review for the positive feedback on the contribution and design on the dataset. as well as the visual and experimental analysis and findings.
> **Due to the page limit, please refer to the *[general response (author rebuttal)](https://openreview.net/forum?id=0hwq2vOHT4&noteId=EtVOyLQxeQ)* for the motivation of DOD task.**
> We address the reviewer's comments below.
>
> ### 1. Comparison with zero-shot grounding.
>
> Thanks for your reminder. Zero-shot grounding is an intriguing study focused on locating concepts absent in the training set. However, it assumes the presence of objects described by query phrases in images (as shown in the Figure 4 of their paper), still falling under the REC task. In contrast, DOD aims to detect objects described by flexible expressions throughout the dataset. Thus, there can be zero, one or multiple objects described by the language reference in an image. The specific differences includes:
> - assuming the existence of objects (zero-shot grounding) vs. no such assumption (DOD).
> - one target only vs. multiple target.
> - short phrases vs. varied language description (from short category name, to phrases, and long descriptions).
>
> DOD and zero-shot grounding have different focus and zero-shot grounding can be regarded as a variant of REC and a subset of DOD. We will incorporate this discussion into the manuscript to avoid potential misunderstandings. Thank you for your suggestions.
>
> ### 2. Motivation for the DOD task. Why not having DOD and REC separated.
>
> As another reviewer also ask about the motivation, we put the clarification on the motivation in the ***[general response (author rebuttal)](https://openreview.net/forum?id=0hwq2vOHT4&noteId=EtVOyLQxeQ)***.
>
> Additionally, regarding the mentioned "oversized glove on left hand", simplifying it to "glove" for object detection is used to highlight the second attribute (unrestricted description) of this dataset. For illustration purposes, only positive examples are included in the figure to avoid introducing information related to the first attribute (complete annotation). In reality, a glove might not necessarily be on the left hand, nor is it guaranteed to be oversized. Thus, the annotation of "oversized glove on left hand" does not align with the label "glove," and there are cases where a positive example of "glove" is a negative example for "oversized glove on left hand". For instance, in supp. Fig. 1 top row, "partially damaged car" cannot be simplified to "car." Applying a "car" object detector would lead to detecting numerous undamaged cars, generating an excessive number of detection results that do not achieve the intended objective.
>
> ### 3. In OFA-DOD which OFA is chosen, base or large?
>
> Thanks for the reminder. We build and evaluate OFA-DOD based on OFA-base. We wil add this note in the text and add the "base" subtext in Tab. 2 in the manuscript.
>
> ### 4. In supp. Table 2, why is Grounding-DINO better than OFA-DOD on Average Recall? Wouldn't one prefer G-DINO in practical case?
>
> We discuss the metric average recall in the supp. Line 180 - 183. Recall is a metric used by REC task, which only requires the model to locate an object known to exist, but no need to reject false positives. A model pursuing high recall can predict as many false targets as it want as long as the ground truth targets are included. Recall is not suitable for detection tasks like OVD or DOD, which requires the model to distinguish and reject negative instances.
>
> In the evaluation of models for DOD, we use mAP to evaluate models' ability to both locate positive instances and reject negative instances. Average Recall is merely a metric for analyzing the characteristics of different models and its value does not reflect the quality or applicability of a model. Actually, we show in Sec. 5.2 of the manuscript that REC or bi-functional methods like G-DINO are difficult to reject false positive instances.
>
> The proposed model is lower on Average Recall compared to G-DINO. This implies it is more "conservative" in prediction and tends to predict instances when it is rather certain. The choice of models depend on the use cases. For most detection settings where false positives are not welcomed, the proposed baseline should be more competitive in performance. When false positives are OK and the user just want to cover as many targets as possible, G-DINO is a nice choice, also considering its wide application and integration available in the community.
>
> ### 5. More discussion on limitation.
>
> Indeed. Thanks for the suggestion. We will add more discussion as below in the "limitation" section of the manuscript:
>
> > As a human-curated dataset, D3 benchmark inevitably contains some bias during data collection and annotation. When designing the dataset, though we try our best to cover as many scenes as possible and make the image distribution and language diversity very broad, the evaluation is still heavily dependent on the choices of language descriptions and the distribution of images.

---

> > ### Comment · Reviewer_SdJt · 2023-08-21
> >
> > Thank you for the response. Additional details on the annotation process are helpful, as well as the added motivation for the task.
> >
> > The visualizations provided in the paper however don't have "oversized" glove, and my example was based on that example. It is unclear how many other such examples are there and perhaps more qualitative analysis could be useful.
> >
> > As such, I keep my score.

---

> > > ### Author Response · Authors · 2023-08-22
> > > **Response to Reviewer SdJt's Feedback**
> > >
> > > Thanks for the feedback.
> > >
> > > The reviewer questioned the annotation of "oversized glove on left hand" (last row) in Fig. 2 of supplementary material is indeed annotation on "glove".
> > >
> > > We want to clarify that the "oversized glove" refers to gloves significantly larger than regular size (i.e. human hand size). As an evidence for this, in the rightmost image of this row, there are two gloves with regular size on both hands of the baseball player on the left. As these two gloves are **not "oversized" and "on left hand" at the same time, they are not annotated. This shows we are actually annotating "oversized glove on left hand" rather than "glove".** Gloves not significantly larger than regular size or gloves on right hand were not annotated.
> > >
> > > In this figure we mainly shows the "unrestricted descriptions" characteristic of the dataset, so we use mostly positively annotated samples of "oversized glove on left hand", which are also positive for "glove". As a better example, we shows some samples of cars damaged (annotated) and not damaged (not annotated) for "partially damaged car" category in the examples in Fig. 1 of supplementary material, which focuses on "complete annotation" (of both positive and negative samples) for the dataset.
> > >
> > > We hope this would clarify the reviewer's question on annotation and we would add more qualitative cases, along with discussions, in the manuscript. Thanks for the suggestions.

---

### Official Review · Reviewer_Q5JG · 2023-07-11

**Soundness:** 1 poor
**Presentation:** 3 good
**Contribution:** 1 poor
**Rating:** 7
**Confidence:** 5

**Summary:**

The paper proposes a task called described object detection which involves detecting objects through free form text queries, encompassing referring expression comprehension as well as open vocabulary detection.

**Strengths:**

## Clarity
The paper is written quite clearly.

## Originality, Quality and Significance: Please see weaknesses

**Weaknesses:**

# Major issues:

## Quality [Annotating using CLIP is insufficient]
* DOD does not provide manually annotated explicit negative certificates for the images that are deemed as a negative for a given text query. The negatives are extracted using a CLIP matching score for each image with all the possible text queries. It has been demonstrated in several works that CLIP has no fine-grained understanding of the image [1,2,3], is largely incapable
of performing spatial reasoning off-the-shelf, obtaining chance performance even on simple synthetic images [1] and also behaves as a bag of words when understanding the text [2]. This implies that any complex query that is more than just a category name, cannot be appropriately distinguished as being relevant or not for a given image, and using this as a step in the annotation pipeline is sure to introduce errors. Using any sort of image-text model during the annotation process inherits the biases of the underlying model and in my opinion is not a viable approach for constructing an evaluation benchmark. For a benchmark that is characterized as a detection dataset, having accurate negatives is paramount, and the authors have not demonstrated that this criteria can be met using a CLIP matching in the annotation process.
* Further, testing the ability of models to distinguish the presence or absence of a textual query, and localize it, would be truly tested in the cases of having unlikely phrases or combinations of objects and attributes or unlikely relations. Using CLIP in the annotation process would completely fail in these cases, as it has been shown that CLIP has a strong Concept Association Bias [3], frequently giving the highest matching score to the most likely completion, without paying attention to the image. This would further exacerbate the difficulty in accurately evaluating models on examples that might be especially hard and interesting (especially relevant to the more challenging "absence" type of queries).

## Originality, Significance [Proposed task is equivalent to existing benchmarks]
* As far as I can tell, the proposed task does not differ from the Phrase Detection task proposed by [5] which addresses the problem of both identifying whether the phrase is relevant to an image and also localizing the phrase, across a whole dataset. Departing from referring expression comprehension, they allow prediction of multiple boxes per phrase, and different from object detection, they evaluate text prompts that are longer than simple category names. Missing this line of literature completely is quite a red flag given that it has been around for quite some years now.
* Another benchmark, "COPS-Ref" has also been proposed [6] that focuses on referring expressions with varying degrees of complexity and in which the localization must be done across multiple images, also containing distractors. This work also does not acknowledge or differentiate from COPS-Ref.


[1] ReCLIP: A Strong Zero-Shot Baseline for Referring Expression Comprehension. Sanjay Subramania et al, 2022

[2] When and why vision-language models behave like bags-of-words, and what to do about it? Mert Yuksekgonul et al, 2023.

[3] When are Lemons Purple? The Concept Association Bias of CLIP. Yutaro Yamada et al, 2022

[4] Revisiting Image-Language Networks for Open-ended Phrase Detection. Plummer et al, 2020

[5] Cops-Ref: A new Dataset and Task on Compositional Referring Expression Comprehension. Zhenfang Chen, 2020

**Questions:**

## Suggestions
* Line 103: "Currently, OFA holds the SOTA among REC methods." I believe this comment is a quite outdated and can be updated (joint OVD & REC methods such as FIBER [1] outperform it).
* For the rebuttal, the authors could explain the difference between the proposed task and phrase detection.

[1]  Coarse-to-Fine Vision-Language Pre-training with Fusion in the Backbone. Dou et al, 2022

**Limitations:**

Yes

---

> ### Author Rebuttal · Authors · 2023-08-10
>
> We thank the reviewer for the feedback. **Due to the page limit, please refer to the *[general response (author rebuttal)](https://openreview.net/forum?id=0hwq2vOHT4&noteId=EtVOyLQxeQ)* for the description of annotation process of D3 dataset.**
>
> ### 1. Annotation using CLIP is insufficient. DOD does not provide manually annotated explicit negative certificates for the images deemed as negative for a text query.
>
> Thanks for the question.
>
> We want to clarify that we do provide **manually annotated negative certificates**. We apologize for not describing the annotation process detailedly in the paper, and have added a diagram to illustrate this process, together with detailed explanation, in the ***[general response (author rebuttal)](https://openreview.net/forum?id=0hwq2vOHT4&noteId=EtVOyLQxeQ)*** above. We hope the reviewer will look into this.
>
> Actually, we provide **negative certificates for all categories except positive categories** on an image. We do not take the federated annotation manner in large-scale, many-classes detection dataset, which only labels partial negative categories for an image due to large annotation cost of negative classes.
>
> In our annotation process, we composite the following behaviors to ensure the negative labels are accurate and not missed:
>
> 1. For the **data source**, the images are divided into different scenarios (groups), and the refs from different group are manually designed to have a small overlapping with each other (i.e., the refs of one group are not likely (but possible) to appear in images from another group).
> 2. For image A1 from group A, we select (1) all the refs from group A, which are likely positive, and (2) partial refs from groups except A, proposed by by top-(n) according to the matching score between the image and refs by **CLIP**. To avoid this operation to filtering out some positive refs that should be kept, we use a rather large value of n (initially 40).
> 3. The **annotators** select 5% images to check that the selected candidate refs covers all positive refs and the filtered out refs are negative. If there is a positive refs filtered out, the value of n is increased to cover that ref, and go back to step 2. After this check, the selected refs includes both positive and negative and the refs filtered out are negative only.
> 4. The **annotators** annotate each image by selecting positive refs from candidate refs and adding boxes. They check that other candidate refs are negative refs.
> 5. The **annotators** check all the images for 2 rounds, the first on all images and the second on 5% images.
>
> With the 5 conditions above, we make sure that for each image, the refs not labeled as positive will be given a manual negative certificate. Such exhaustive and complete annotation is possible by limiting the scale to be 10000+ only, as an evaluation benchmark, and utilizing CLIP. But we do not rely on CLIP for deciding a category is positive or negative for an image.
>
> ### 2. Using CLIP in the annotation process would fail in cases of having unlikely phrases, attributes or relations.
>
> Thanks for the insightful question. As shown in the annotation process (diagram and text) and the answer #1, in the annotation process of D3, CLIP merely provides some initial candidate refs. Since
>
> (1) refs likely to be positive are bound with the image's group and always kept as candidates,
> (2) the selection percentage of CLIP is large (>10%) and adjusted based on manual check,
> (3) the refs not selected by CLIP is manually checked by annotators to be negative,
> (4) the annotators decide a ref is positive or negative,
> (5) the final annotations are checked by annotators twice,
>
> we argue that the proposed D3 dataset offers manually labeled, accurate negative and positive labels with explicit negative and positive certificates, and CLIP only serves as a tool for accelerating the annotation process without deciding the positive/negative or harming the annotation accuracy.
>
> ### 3. Difference with Phrase Detection task and existing benchmark COPS-Ref.
>
> Apologies for leaving out these two relevant works. The main difference between DOD and Phrase Detection [4] is that Phrase Detection lacks explicit negative certificates. Negative instances are not labeled, so Phrase Detection is not a detection task. In DOD, we ensure that positive instances are annotated exhaustively and all the other references are reliably negative labels. Additionally, Phrase Detection focuses solely on the form of phrases, while DOD encompasses OVD and REC, allowing expressions to be words, phrases, or even sentences.
>
> Cops-Ref [5] focuses on assessing the grounding capability of the REC method in difficult negative regions with related/distracting targets, ensuring explicit negative certificates for a small set of images in their benchmark. Thus, achieving the 'explicit negative certificates across a whole dataset' attribute, like in detection tasks, is only feasible in DOD.
>
> We will include this discussion in the manuscript. Thanks for your valuable suggestions.
>
> ### 4. Update REC SOTAs like FIBER. OFA is not SOTA.
>
> Thanks for the reminder. In this work, we divide existing methods into REC, OVD and bi-functional methods. As FIBER handles both detection and REC, we believe it is more suitable to be classified as bi-functional methods rather than REC methods. Therefore, we think the expression "OFA holds the SOTA among REC methods" is valid. We also make comparison to joint OVD & REC methods like UNINEXT (outperforming FIBER on most metrics of REC) and Grounding-DINO (comparable to FIBER) in the paper.
>
> We will add methods like FIBER in bi-functional methods and MDETR in REC methods in the related work of the manuscript. Thanks for your suggestions.

---

> > ### Comment · Reviewer_Q5JG · 2023-08-18
> > **Response to rebuttal**
> >
> > Thanks for the clarifications and for the detailed response to my questions. I have one follow up question after which I would be happy to raise my score. Reviewer YHXi brought up the point about evaluation on phrases that may be contained within a longer phrase (such as "backpack" and "yellow backpack"). Could you please clarify how this is handled by clearly explaining how the mAP metric is calculated in these cases or in the case of hypernyms like boat / canoe?

---

> > > ### Author Response · Authors · 2023-08-18
> > > **Response to Reviewer Q5JG' Feedback**
> > >
> > > Thank you very much for reviewing our responses. We are grateful for the feedback! We address the new query with the following points:
> > >
> > > 1. **Regarding category relationships, parent-child or synonym are avoided in design, but partial overlap is acceptable**. When designing the categories, we intentionally avoided incorporating such parent-child relationships between categories, and also synonym relationships, to ensure greater diversity and challenge within the dataset. However, there is some partial overlap between categories. For example, "dog not lead by rope outside" and "clothed dog" do not have a parent-child relationship but can overlap in certain cases.
> > > 2. **Detection on $D^3$ is multi-label, making it suitable for categories with relationships**. Considering possible relationship between categories, detection on $D^3$ is multi-label rather than single-label. An effective detector should assign all relevant positive categories (e.g., "dog not lead by rope outside" and "clothed dog" for a clothed dog not lead by rope outside) for an instance.
> > > 3. Given the multi-label setting, **our exhaustively labeled dataset does not require a specially designed metric for category relationships.** In $D^3$, as all positive and negative labels are known for an instance, the relationships between different categories will not affect the evaluation, so we can use consistent evaluation for each category across all images. Comparatively, for datasets like LVIS with non-exhaustive federated labeling, when relationships between categories exists, the partial labels can introduce errors on unknown categories, so such categories may need to be handled specially. $D^3$  is not susceptible to this issue.
> > > 4. **We use the standard detection mAP as the evaluation metric.** The evaluation is similar to COCO mAP and we base its implementation (will be open-sourced) on `pycocotools` . For inference, an instance predicated with category A and B is regarded as an instance for category A and an instance for B. The AP for each category is computed as follows: *Predictions for each category across all images are sorted by score in descending order, and those with a ground truth IoU exceeding a threshold are counted as TP (and the ground truth is marked as taken), while the rest are counted as false positives.* With these TP and FP instances, we calculate the precision, recall, and AP following COCO. The mAP is calculated by *averaging the AP across all categories*.
> > >
> > > In conclusion, the exhaustive annotation in our dataset, unlike federated datasets such as LVIS, and the multi-label setting, which accommodates categories with relationships, ensures that direct AP evaluation for each category is suitable and does not introduce errors. This is attributed to our dedicated design of dataset metadata and annotation process. The standard mAP metric adopted shows our dataset adheres to the stringent requirements of a standard detection dataset and meets the demands of the DOD task. If the reviewer has any further inquiries, we will be happy to answer. Thanks for your help in making this work better.

---

> > > > ### Comment · Reviewer_Q5JG · 2023-08-19
> > > > **Thanks for the details!**
> > > >
> > > > This sounds good to me! Thanks for the great new dataset! If I may suggest one final thing - I think this paper would benefit from having a more descriptive title. The current one throws the reader off because it's hard to know what the paper is about (since "exposing the troublemakers" doesn't provide any information, and described object detection is not a standard term as far as I know). Maybe consider changing it to be more descriptive of the dataset you are proposing so it has more impact!

---

> > > > > ### Author Response · Authors · 2023-08-19
> > > > > **Thanks for your review!**
> > > > >
> > > > > Thank you sincerely for your invaluable feedback and comprehensive response. Your review contributes to the enhancement of this work. We wholeheartedly appreciate your insights and will seamlessly integrate our discussion with the existing materials.
> > > > >
> > > > > In regard to the title of the paper, we are fully aligned with your perspective. Your suggestion resonates with us, and we acknowledge that the current title is not clear and informative enough. We are firmly committed to revising the title to be more descriptive of the proposed dataset.

---

### Author Rebuttal · Authors · 2023-08-10

We thank the reviewers (R1: Q5jG, R2: SdJt, R3: EwWL, R4: YHXi, R5: wkHv) for their positive feedbacks, such as the contribution of the dataset (R2, R5), the significance of the target problem (R3, R5), the design of the dataset is throughly considered (R2), the absence description characteristics is interesting (R2, R5), the findings in the comparison of different tasks is informative (R2, R4, R5), the experimental and visual analysis is interesting and comprehensive (R2, R3, R4, R5), the proposed baseline is effective (R2, R3), the writing is clear and clarified (R1).

In the general response we answer some questions asked by more than 1 reviewer. We address each reviewer's question in the individual response for them. We will revise the paper accordingly.

## Annotation process of the proposed $D^3$ dataset.

A diagram illustrating the annotation process of the proposed dataset is in the PDF file. Here we describe the steps for annotation as below:

Data source: 106 groups from GRD with about 100 images and 3 ~ 4 designed refs for each group. Each group belongs to a different scenario and the overlapping between refs from different groups are small (i.e, a ref for one group are not likely (but possible) to appear in the image from another group). Now we have 10000+ images and 300+ refs.

1. [Manual] Adding absence refs: design 1 ~ 2 absence refs based on the images for each group and add them to the corresponding groups. Now we have 400+ refs.
2. [Automatic] Selecting possible positive refs: for each image, select **all the refs **(4 ~ 6) from the group it belongs to, and also the other 105 groups (top-n refs out of 400+ refs, by CLIP similarity between the image and each description). Now for each image, we have n+4 ~ n+6 candidate refs and all the other refs are filtered out. n is set as 40 initially.
3. [Manual] Verification: randomly choose 5 groups of images, and check if there are any positive refs that should not be filtered out. If so, increase n to cover that ref and go back to step 2.
4. [Manual] Manual annotation: annotation by trained annotators on all images. The annotation of boxes (and instance masks) are instance-level, dataset-wise complete, and includes absence refs.
5. [Manual] Quality check: this includes 3 small steps:
   1. Discarding some images (ambiguous, etc., unsuitable for annotation) or categories from the dataset. About 8% samples are discarded.
   2. Quality check on 100% samples. For each group, if image with error is more than 2%, it is returned for re-annotation. Otherwise the errors are fixed and this group passes this step.
   3. Final check on 5% samples. For each group, if there are image with error, it is returned, otherwise it is accepted.

## Motivation of the proposed DOD task.

Both the OVD and REC tasks have their respective limitations.

- OVD can only perform detection based on categories, where the detection targets are limited to "certain object classes" rather than "objects with specific attributes/relationships." This approach lacks an understanding of contextual information within images and cannot leverage language to precisely control detection targets and requirements. This inflexibility prevents it from meeting specific application demands.
- REC, while capable of comprehending longer object descriptions with attributes or relationships, assumes the existence of such objects in the image. In cases where the described object doesn't exist, REC lacks the ability to reject or filter, leading to false positive errors. This issue poses a significant problem for practical applications and limits its direct usability.

Consider a practical scenario, such as detecting "individuals without helmets" in a construction site using camera data. An OVD method can detect objects like "helmets" and "people" and generalize, but it can't determine the relationship between people and helmets, rendering it unsuitable for direct application. On the other hand, the REC method produces localization results in any image, but often generates false positives, making it impractical.
The current solutions involve breaking down the process, first detecting "people" and "helmets," then training a separate model to determine the relationship between them, or determining presence first, followed by the REC method for localization. This approach requires multiple specialized models, tailored for each scenario, which is far from practical and is inefficient in terms of development.

Hence, there is a significant demand for detection based on language descriptions – a model with strong generalization capabilities, capable of determining whether the described object exists in the image and localizing it based on arbitrary language descriptions. This is where our proposed DOD task comes in.
The introduced DOD task has various practical applications, including:

- Urban security, like detecting "individuals without helmets" in construction sites, "dog outside without leash" in communities, "clothes hung outdoors" on a street, "overloaded vehicles" and "fallen trees on roadsides" on the road, etc.
- Network security, where sensitive images containing bloodshed or violence need to be detected within a massive image dataset.
- (Fine-grained) photo album retrieval based on language (descriptions, keywords, etc.).
- Retrieval and filtering of web image data.
- Detection of specific events in autonomous driving, such as "pedestrians crossing the road".

These scenarios are beyond the capabilities of both OVD and REC. This is the motivation behind DOD.

## Content of the attached PDF

In this PDF file we include 3 additional figures:
- Figure 1 is the diagram of the annotation process for $D^3$.
- Figure 2 and Figure 3 show the model structures of OFA and the proposed OFA-DOD.

---

### Decision · Program_Chairs · 2023-09-21

**Decision:**

Accept (poster)

**Comment:**

This paper extends open-vocabulary object detection to support short and long-form descriptions rather than short category names, referred to as Described Object Detection (DOD). To address this, a (evaluation-only) dataset and benchmark is proposed, which is fully annotated and includes a variety of description characteristics. Benchmarks show that methods developed for close settings (open-vocabulary detection and referring expression comprehension (REC)) do not perform well. Some simple modifications are proposed to improve results, including a dual-decoder and reconstructed data, although the task is still challenging for modern methods.

  While the reviewers appreciated the setting and clear writing, a number of concerns were raised including: 1) Dataset generation, e.g. using CLIP (Q5JG, wkHv), 2) Setting design, e.g. defining categories and evaluation metrics (YHXi), 3) novelty and motivation of the setting (Q5JG, SdJt), and 4) assessment of related methods such as OVD methods (EwWL). The authors provided a rigorous rebuttal including a thorough description of the annotation process and motivation, additional experiments (OFA-DOD and fine-tuning), and more.

  By the end of a rigorous back-and-forth, all but one reviewer suggested acceptance and were satisfied with the response, either keeping their positive score or increasing it. For the reviewer that had a more negative score, there again was a great discussion of potential limitations in terms of overlap between the base/novel classes, evaluation of both novel and base class performance and use of mAP for this, etc. While not all points were resolved, I believe that a majority of them were and on the whole, this paper presents a valuable, impactful, and scientifically interesting setting that could form the foundation of interesting lines of research. It is also well-executed, with thorough analysis and insights, which all of the reviewers agreed.

  I therefore recommend acceptance, but heavily encourage the authors to incorporate the many additions and perspectives offered by the reviewers. This will significantly increase the impact of the paper.